# The Simpsons did it: Exploring the film trope space and its large scale structure

**Pablo García-Sánchez** [1]*, **Antonio Velez-Estevez** [2], **Juan Julián Merelo**[1], **Manuel Jesús Cobo**[2]

**1** Department of Computer Architecture and Technology, University of Granada, Granada, Spain,
**2** Department of Computer Science and Engineering, University of Cádiz, Cádiz, Spain

* pablogarcia@ugr.es

**Data Availability Statement:** Dataset is available at https://github.com/raiben/made_recommender/blob/master/datasets/extended_dataset.csv.bz2.

**Funding:** This work has been partially funded by projects DeepBio (TIN2017-85727-C4-2-P, lead by JJM) (Ministerio de Economía, Industria y

## Abstract

Creating a story is a challenging task due to the the complex relations between the parts that make it up, which is why many new stories are built on those cohesive elements or patterns, called *tropes* that have been shown to work in the past. A trope is a recurring storytelling device or pattern, or sometimes a meta-element, used by the authors to express ideas that the audience can recognize or relate to, such as the *Hero's Journey*. Discovering tropes and how they cluster in popular works and doing it at scale to generate new plots may benefit writers; in this paper, we analyze them and use a principled procedure to identify trope combinations, or communities, that could possible be successful. The degree of development of these different communities can help us identify areas that are under-developed and, thus, susceptible to such a type of development. To detect these communities, with their associated degree of development and interest, we propose a methodology based on scientometric and complex network analysis techniques. As a secondary objective, we will obtain a general perspective in the trope and films network: the *tropesphere*. We have used a dataset of 10,766 movies and 25,776 tropes associated with them, together with rating, genres and popularity. Our analysis has shown that not only there are different trope communities associated with specific genres, and that there are significant differences between the rating and popularity of these communities but also there are differences on the level of development between them: emerging/declining, specific, transversal or motor.

## 1 Introduction

Finding inspiration in creating new stories is a struggle for many creators. Drawing creativity from stories and ideas that have been well received in the past can not only help professionals, such as filmmakers and writers, to create interesting (and profitable) works, but using data from past stories and ideas can also be used as input for automatic story generators. Increasingly, narrative creation is data driven, with streaming platforms basing their decisions to produce new works on existing data [1]. In many cases, new platforms like videogames simply lack the tools to create massive backstories for non-playing characters, so they are simply not developed [2]. However, outside proprietary platforms, there are multiple ways to collect data

Competitividad, Gobierno de España (ES)) and project iScience (PID2019-105381GA-I00, leaded by MJC) (Ministerio español de Ciencia e Innovacíon). The funders had no role in study design, data collection and analysis, decision to publish, or preparation of the manuscript.

**Competing interests:** The authors have declared that no competing interests exist.

on released works, and analyzing this information will opens a window towards understanding how humans consume and interact with different cultural media, making some of these cultural works more popular, or simply better, than others.

Every type of media has different languages and means of expression. However, underlying it all is plot, its structure and patterns that are repeated in it, independently of the final expression. Professional script writers and storytellers are familiar with the concept of *trope*. A trope is a recurring narrative pattern or device that appear in cultural works. One of the most famous examples is the *Hero's Journey* [3]: the protagonist is called to adventure, an old mentor hands him a weapon, meets allies, confronts evil, and returns home as a hero. *Willow*, *Star Wars: A New Hope*, *The Hobbit* or *Harry Potter* are clear examples of movies that use this trope, although this trope is as old as the *Homer's Odyssey* or even older. Other famous ones are *Chekhov's Gun* (a device is presented early in the film to be used later), the *Mexican Standoff* (several characters pointing their guns at each other) or the *McGuffin* (an object, that could be magical, that drives the plot). Tropes are intentionally used in different media, such as films, books, video games or comics to obtain surprising or interesting effects, based on a certain familiarity or recognition capacity by the audience; in other cases, they simply are identified after they show up in different pieces. It is virtually impossible to create a story without tropes. In fact, "The Simpsons did it" is a catchphrase used to indicate that an idea that seems to be new or original has already been used in an episode of the animated series *The Simpsons* [4] that run for more than twenty seasons. However, it is obvious that tropes can be created, they are not simply *there* to be picked up by authors; they can also be combined or laid over a plot in totally original forms. Any of these could be a factor in creating a popular, or simply well-rounded and finished, work. Researching the use of tropes in different cultural media can be very interesting in different fields. For example, it is possible to predict the note of a film from the tropes that form it [5], or to create conceptual spaces between them to help on their study [6].

Although tropes cannot be used to fully model the plot of an artistic work, they can be considered as the narrative architecture on which stories are built, and therefore, it is possible to characterize a work (a film, a comic) from the tropes that form it [5]. In fact, writers do not have to use it consciously, but sometimes tropes emerge from the structure of the plot. Moreover, there are even tropes not related to the narrative itself, such as the *Shout Out* trope, in which a reference or joke is made to another external work.

One of the most important sources of tropes in different media is the TV Tropes wiki (https://tvtropes.org/); created and curated, in principle, by anyone interested, this web site currently contains descriptions of nearly 30,000 tropes that appear in more than 80,000 works in different media, such as anime, cinema, literature, television shows or video games, among others. The semi-structured text, links and tags constitute a large dataset that has been generated by *crowdsourcing*, that is, it is open to anyone to contribute, by adding or editing the information, so the data collected may vary over time, or even be biased towards the most popular preferences at the moment. In the work by García et al. [5], the authors scraped the dataset of TV Tropes films along with its associated trope list, and mapped it with the IMDb website to obtain the rating that the users of that website have given to each film. From this dataset they trained a neural network, which served as a surrogate model for a genetic algorithm that optimizes a fixed-size set of tropes to maximize the possible score. Although they performed a preliminary analysis of TV Tropes, no communities detection study of the complex network provided was performed, nor was there any relationship between trope communities, genres, and rating and popularity. In fact, the analysis of information structured as complex networks can give us information about how humans interact and consume cultural media [7].

The aim of this paper is to find out how tropes affect the rating and popularity of the films in which they appear, as well as what is the degree of development of the different communities of tropes: groups of tropes that are cohesively used in many films and therefore are more mature, or on the contrary, cohesive tropes used in only specific films, but also, we can detect which communities are more marginal, and are therefore, emerging or decaying. This will allow us to find which areas of the *troposphere* (the complex network that models the relationships between the tropes from the films in which they appear) can be an interesting niche for developing new films or new linear narratives in any other area such as games. In this endeavor we should not lose sight of the fact that the source that we are using is biased towards popular and recent films, so some stories that have not reached popularity for some reason (minority language, for instance) or that were released a long time ago might have a winning trope combination that is not reflected in our study. Besides, this bias is not constant, so we cannot really affirm that some specific genre or kind of movie has been left out (or included) uniformly across the data we have analyzed.

Fortunately, this bias aligns with our objective, which is narrative generation in the present sense, as well as the creation of narratives that might be cohesive. We will assume that a part of what makes a movie popular is the cohesiveness of the tropes they employ, or how they work together with each other. This means that our results will probably not be affected by this bias, that will nonetheless have to be taken into account when working towards a different objective.

As a secondary objective, this paper proposes a methodology that allows us to find combinations of tropes that should appear in a film in order to be popular, but also, it can give us a vision of the current state of the troposphere, that will allow us to know how the audience consume and interact with such an important cultural medium as films.

The methodology proposed in this paper is based on the analysis of co-word networks in scientometrics. In this field, networks of co-words are made from the keywords associated by hand with the paper by its authors (approximately 3 to 5 per paper) [8, 9]. From this co-word network, communities can be obtained using algorithms such as the Leiden Algorithm [10]. The impact of these communities can be measured by comparing the average number of citations or the H-index of the papers belonging to each community. Keyword overlapping analyses are also often performed, for example, by dividing the papers by the degree of international collaboration (local, national or international) to obtain relevant information [11]. The degree of development of each community is performed by Strategic Diagrams [12]. Applying this methodology to the tropes that appear in the films involves some differences. The main difference is that not all films have a homogeneous number of tropes as papers do, either because of the characteristics of the film itself or because they have not received the same attention from users when generating the dataset. Therefore it is necessary to perform a descriptive analysis of the dataset to see if there are any biases. On the other hand, measuring impact is more complicated, since unlike the average number of citations, metrics such as interest (number of votes) and average score are subjective, and the results should be taken with a grain of salt. In fact, just as using a particular keyword does not guarantee more citations in a paper, the impact of a keyword/trope is an indicative of the attention given to a particular theme, not that its mere use guarantees a benefit. Other factors need to be considered, such as the quality of the results in the case of the papers, or extra information related to the films, such as directors, actors or advertising investment, among others. Finally, a keyword defines a paper better than a trope defines a movie because of its more abstract nature. For example the abstract *Chase* trope can be performed by car, ship or running, among many others, and during different settings or time periods.

However, this methodology can provide a certain degree of utility, for example in finding which combinations of tropes and their degree of development can be interesting when creating new films, or at least a pitch for a new one, or to obtain a snapshot of the current state of the tropes in a specific medium (movies in this case). Due to the different issues we want to cover in this paper, we have decided to address the following research questions (RQs):

- RQ1: *Are there any biases in the films or in the tropes available on the dataset?* Since we are going to use a large dataset created through crowdsourcing, unlike data obtained from bibliometric data sources, it is necessary to understand how the information in this dataset is distributed before making any assumptions after analyzing it. For example, find out which genres or films receive the most attention from dataset contributors. These biases will indicate how well the set of tropes detected by the users will match the *actual* set of tropes in a movie.

- RQ2: *Is there any overlap between the tropes of different genres?* Since genres are a primary and well-known classification of films, understanding which tropes belong to which genres will also give us insight into the results, and will also help us understand differences between the different communities of tropes obtained in the following steps. This will also help us shape or constrain the tropes that are going to be used to generate a story; generally genre (or combination thereof) comes before the plot itself, and understanding how tropes shape genres or the other way out will help to generate better stories.

- RQ3: *Do sets of related tropes influence the rating and popularity of a film?* Once we have obtained the different communities of related tropes, we can measure the interest of the films that form them, being able to compare the differences between these communities. We are assuming that at least a big part of the popularity of a film comes from the specific tropes, or combination of them, that are being used. We will try and find out to what extent this happens.

- RQ4: *Is there a difference in the degree of development of the thematic of the different trope communities?* By studying the centrality and density of the different communities obtained, it is possible to visualize their relationship between the tropes that form each community, or how these communities relate to each other. This will allow us to measure how *motor*, *transversal*, *specialized* or *emerging* these communities are [9], and eventually identify which areas of the troposphere deserve to be developed, that is, which specific tropes could be used and how they could be mixed together to create popular films.

To answer these questions, we will apply this methodology to conduct a study on the relationship between tropes, genres and the rating and popularity of the films that use them. To do this, we will perform an overlapping analysis to see which tropes appear in which genres, as well as a network of co-occurrence of co-films and co-tropes. We will apply a community detection algorithm on these complex networks and compare and visualize their metrics to extract the required information.

This methodology will be applied over a dataset of films, tropes, years, genres, rating and popularity. We will download the films and tropes available at the TV Tropes website, following the process explained in [5]. Then, we will map these films to the ones in IMDb in order to extract complementary information. Finally, we will perform different analysis and visualization methods to that dataset.

Moreover, understanding how the use of tropes has evolved throughout the history of a specific artistic expression area can give us information about the cultural and social changes of that medium. Therefore, our methodology can also help academics on different fields, such as

social sciences, to extract knowledge, but also allow to storytellers adapt to these changes or even extract new ideas.

In general, we think that this methodology can be useful since it gives us a quantitative approach to the analysis of narrative devices, which can be easily turned into a methodology for generation of narratives, as a well as a for critical analysis of these systematized narratives in the past. In this sense, we think it can benefit video game creators, as well as media history scholars.

The rest of the paper is structured as follows: first, the state of the art in trope analysis is discussed. Next, the methodology used to obtain the dataset and the different metrics studied is described. In Section 4 the experimental results are discussed, and finally the conclusions and future work are presented in the Section 5.

## 2 Background

The creation of film scripts is a very challenging task due to the complexities of the plot, since the elements that are part of a work must fit well, because a small failure can cause a cascading disaster [13]. In fact, the concept of narrative can be seen as a complex adaptive system, in which the interactions of its elements and events make the story emerge [14].

As stated in the introduction, tropes are the literary constructs that define the overall narrative of a work. While they do not define the plot unequivocally, they constraint it in a number of ways.

Actually, the concept of "genre" of a film (Thriller, Horror, Comedy, Western, etc.) and the "trope" are very aligned and interrelated, and both help on describing a film from different perspectives: whereas a genre relates to the type of the story from a high-level point of view, a trope is a tool or device to achieve a specific effect on the story and can vary from high-level structures to details. The concept of "Genre" is an important source of discussion of film theory, and there is some controversy about the definition of this concept, and how to classify films. In fact, they are easier to recognize than to define, and scholars agree that they cannot be rigidly identified [15, 16]. There are different distinctions between genres, for example, the universal ones (Action, Comedy, Drama, Horror, Mystery, Romance and Thriller) versus the setting genres (Western, War, Sci-Fi or Musical, among others). Classifying a film into a single genre is also difficult, since, for example, a science fiction film such as *Back To The Future III* can also be classified within the Western genre.

There are available in the literature some works studying specific tropes in film and media studies. For example in [17] action films tropes are discussed, including characters tropes (cop, cowboy), actions (chase, last minute rescue) and techniques (for example, camera shaking movement). More works have focused in other kind of tropes, such as the trope of the *Zombies* [18], scenes that take place in a museum [19], or even as specific as the Brazilian housemaid [20].

But as previously stated, and because narrative can be seen as a complex adaptive system, methods such as complex network analysis can be useful for researchers in this field. For example, bibliometric networks [8] have been extensively studied using techniques such as co-word network analysis or strategic diagrams [9]. These methods normally are used to detect communities between keywords of the published papers and to visualize them [21]. Moreover, word analysis has been applied in other contexts such as traditional text analysis. For instance, Herrera et al. [22] proposed a detector, which is based on unsupervised statistical methods, for detecting keywords in texts. Other application to traditional text analysis is presented by Tohalino et al. [23], in which a multilayer network is used to address the extractive multidocument

summarization task. However, in our study we do not analyze corpus of text, but the relationship between movies and the tropes they appear in them, and vice versa.

However, there is a lack of works dealing with complex network analysis in the field of creative media such as films. In [7], authors use a complex network of interactions between literary characters (specifically, the Harry Potter saga) to compute their relationships to see how they evolve throughout their story. Using text extraction algorithms from novels, they generate different social networks from the dialogues. These networks are used to measure some characteristics related to the author's style, and to the book's story. Amancio [24] proposed the use of co-occurrence networks to study and detect the different entities (characters or locations) and their semantic relationships, which appear in different novels. A topological analysis was applied to obtain patterns that were not possible to detect using classical methods. Another application of the use of complex networks is the identification of the meaning of words with multiple meanings. The work [25] describes a process based on a bi-partite network model that outperforms widely used machine learning methods to deal with this issue. However, as stated before, these works rely on the analysis of whole texts to obtain the relationships between entities, whereas our work is based on the mere occurrence of tropes in the different films.

Another example of complex networks are those generated by crowd science movements [26], such as TV Tropes. TV Tropes is a wiki community created in 2004 that unites enthusiasts who have been collecting tropes collaboratively since 2014. It currently has over 28,000 pages of tropes, and 77,000 of cultural works, including films, TV, comics, video games, books and other media. In [26], this wiki community was studied, indicating that the produced network generates knowledge relevant to social and humanities disciplines, and that it can serve as a starting point for academic studies. However, they conclude that the definition of tropes is a relatively complex task, and that the contributors of this community approach it using a qualitative and conceptual approach. On the other hand, and as discussed in [5], being a dynamic, extensive and organic study, a certain type of bias may exist: the most popular and recent films usually receive more attention.

Other authors have used examples and data from this website to model a geometric conceptual space between tropes [6]. Although their study is limited to three tropes –hero, anti-hero and villain–, the fact that a measurable distance between tropes can be modelled may be useful for future research.

To our knowledge, the only work that applies automatic analysis or machine learning techniques to the set of TV Tropes is the one published by García-Ortega et al. [5] whose objective is to optimize which tropes a film should have in order to obtain the highest possible rating.

## 3 Methodology

In order to carry out the analysis, a specific methodology has been designed. It comprises 8 steps which are explained in what follows.

1. *Data acquisition*: We have used the Python package `tropescraper`, available at https://github.com/raiben/made_recommender, used in [5] to obtain all the tropes available in the web TV Tropes.org. For each film available on the web, a list of tropes has been obtained. This package iterates over the tropes of the TV Tropes webpage, extracting all the films associated with a trope. Then a dictionary of films and its associated tropes is created. Next, a mapping process is carried out between the films and the data available on the IMDb website (https://datasets.imdbws.com/), in order to add information such as the average rating, the number of user votes (popularity) and the year. Because the identifier of a film in TV Tropes is only the name, and sometimes the year, in case of ambiguous associations we

have performed the match taking into account the popularity in the IMDb dataset. The rest of the details of how these datasets can be obtained and how the mapping process is done can be consulted at [5].

2. *Preliminary analysis of the dataset*: To get an overview of the dataset, a descriptive analysis is proposed. The analysis includes the standard deviation, mean, minimum, maximum and the first, second and third quartiles of votes, ratings and tropes. Moreover, a visualization of the distributions of the number of tropes, ratings and votes is shown. Furthermore, a line chart with the films per year is presented. Finally, the votes and ratings in periods of two decades from 1880 to 2020 are analysed as well as the distribution of genres in the dataset. All the previous analysis will help us to detect whether some biases exist or not. Additionally, the analysis will give information to answer the first research question (RQ1).

3. *Dataset pre-processing*: in this study, two datasets are considered:

   - The dataset obtained from the data acquisition phase. In the rest of the text it will be referred as $D_f$.

   - A second dataset, obtained from $D_f$, in which for each trope a list of films which have that trope is obtained. After that, the mean of the votes and rating of the films that have each trope are calculated. These two factors are calculated in order to quantify the quality of each trope in terms of the votes and rating of the films in which it appears. In the rest of the paper, it is going to be named as $D_t$.
   This transformation of the dataset will allow discovering how tropes are related among them based on the films they appear and, inversely, the relations of the films based on the tropes.

4. *Overlapping analysis*. Aiming to know how the genres differ from each other in terms of their tropes (RQ2), the $D_t$ dataset is used to do an overlapping analysis. An overlapping analysis is a process that reveals how two sets are mixed together with respect to the number of elements that both sets have. A well-known measure of overlapping is the Jaccard index [27], which measures the similarity between two sets, and it is defined as the size of the intersection divided by the size of the union of the sets given. In this paper, the Jaccard index [27] is computed for each possible pair of set of tropes, each of them corresponding to a genre. It is is computed from two distinct perspectives, as it is also performed in a similar way in bibliometric studies [11]:

   (a). For all the tropes in genres. This offers a general overview of the mixing between genres.

   (b). For the 100 most voted or rated tropes in genres. This shows how the most relevant tropes in each genre relate to other genres.
   Formally $J_N$ is defined as follows:

   $$J_N(A, B) = \frac{|A_N \cap B_N|}{|A_N \cup B_N|}$$

   Being $A_N$, $B_N$ the subset of the $N$ most valued or rated tropes of each set. If $N = \infty$ all the elements in the sets will be considered, for example, the first perspective we proposed above.

5. *Co-films and co-tropes network construction*. In bibliometrics [8, 12, 28], a co-occurrence network is a graph where the nodes are some unit of analysis (e.g words in co-words networks) and the edges are the co-occurrence relationships between them. That is, there is an

edge between two occurrences if they both appear together in a set of documents. There-fore, co-occurrence networks are built from a set of documents $D$ in which each one has a set of unit of analysis $U$. The process of building the network, called in this paper the *trope-sphere*, is shown in the Algorithm 1.

**Algorithm 1**: Algorithm used to build a co-occurrence network

```
Result: G: The co-occurence network.
foreach document d in D do
  U = get unit of analysis from d
  for i ← 1 to |U| do
    if U[i] not exists in G then
      Add node U[i] to G.
    end
    for j ← i + 1 to |U| do
      if U[j] not exists in G then
        Add node U[j] to G.
      end
      if {U[i], U[j]} not exists in G then
        Add undirected edge {U[i], U[j]} to G with weight 1.
      else
        Increase weight of {U[i], U[j]} edge by 1.
      end
    end
  end
end
```

Following the instructions pointed above, from the original dataset $D_f$ the co-tropes network (tropesphere) is constructed and, similarly, for the $D_t$ dataset co-films network is constructed. The edges were normalized using the equivalence index [9]. To be clearer, in these co-occurrence networks the co-occurrence is measured in two ways:

- Co-films network, the co-occurrence is measured as the number of times that two films have the same trope.

- Co-tropes network, the co-occurrence is measured as the number of times that two tropes appear in the same film.

6. *Community detection*: To determine the groups of films that are related among them and the groups of tropes that are related, for each dataset a community detection algorithm will be applied. Specifically, the Leiden algorithm [10], which is a modularity optimization-based algorithm, will be used since it has demonstrated good behaviour with co-occurrence networks [11]. The modularity is a very usual metric for measuring the quality of the detected communities within the network [29].

7. *Performance measures*: for each co-tropes network community, films belonging to the nodes within the community will be associated by means of the algebraic union. This means that a film can belong to more than one node for the same community. From these communities, different metrics can be obtained, such as the number of films in a commu-nity, the average score, the popularity or the percentage of films of each genre within the community etc. Regarding the co-films network, the metrics are also obtained by the films in the communities (in this case, no films are repeated between communities).
To confirm that there is a statistical significance relationship between the rating and popu-larity of the obtained communities, a normality study will be conducted to determine the most appropriate test [30] to apply. Once this is done, we can answer RQ3.

8. *Visualization*:

The *strategic diagram* is widely used in scientometrics [9, 12, 31], helping visualize the degree of development of a thematic community (cluster) by analyzing the networks of co-words using the keywords of the papers. It is based on two measures:

- Callon's centrality: it indicates how well a theme is connected to other themes (external cohesion). It is defined as: $c = 10 \times \sum e_{kh}$ with $k$ word belonging to the theme and $h$ word belonging to another themes [9, 31].

- Callon's density: it measures the internal strength of a network and it can be defined as $d = 100(\sum e_{ij}/w)$, with $i$ and $j$ belonging to the theme and $w$ being the total number of words in the theme [9, 31].

Moreover, the themes plotted in the strategic diagram are labelled by its central node (referred in the text as *central trope* or *central film*, depending on the network), which is the node with the most degree in the theme, formally $n = \text{argmax}_v(\deg(v))$. Using the callon's density and centrality it is possible to visualize the communities in four quadrants, whose center is in the position (0.5,0.5). These quadrants indicate if communities are motor (central and developed communities), transversal (basic and general), specialized/peripheral, or emerging/declining [9]. This will help to answer RQ4.

Moreover, the structure of networks can be obtained and simplified by the maximum spanning tree of the network. This structure will be represented using the Gephi software [32]. This will make it easier to see which tropes or films are most related to each other, and to see how these communities are organized.

## 4 Experiments and results

In this section, the steps described in the methodology will be applied and the results will be discussed. Since the aim of the paper is to find those areas of the troposphere that may be interesting for the development of new films, as well as to get an overview of the state of the troposphere, we will apply the steps of the methodology on an existing dataset. First we will study the state of the dataset with respect to bias and genres (RQ1 and RQ2). After that, we will analyse the co-word networks (co-tropes and co-films) using communities detection, to detect the differences in popularity and rating between the trope and film communities (RQ3). Finally we will find those thematic areas in which it could be more interesting for the creation of new works (RQ4).

### 4.1 Descriptive analysis of the dataset

As we are using a dataset produced from crowdfunding it is necessary to see if there are any biases towards the most popular or recent movies. To solve RQ1 we will perform a descriptive analysis.

Once the dataset has been obtained from https://github.com/raiben/made_recommender/blob/master/datasets/extended_dataset.csv.bz2 following the steps described in previous section (and in higher detail in [5, 33]) this descriptive analysis of the dataset is summarized in Table 1. The average number of tropes per film is 48.42. As it can be seen, there is a high standard deviation in the number of votes and tropes per film, so a more detailed analysis is necessary in order to study the distribution of the values.

Fig 1 shows the distribution of tropes by film. As expected, the number of tropes by film does not follow a normal distribution, but a log-normal one. There exist a smaller number of

**Table 1. Descriptive analysis of the dataset variables.**

|      | #Votes | Rating | #Tropes (per film) |
|------|--------|--------|--------------------|
| mean | 49306.22 | 6.36 | 48.42 |
| std | 119032.00 | 1.33 | 59.86 |
| min | 0.00 | 0.00 | 0.00 |
| 25% | 2249.25 | 5.80 | 16.00 |
| 50% | 9368.00 | 6.60 | 29.00 |
| 75% | 42095.00 | 7.30 | 56.00 |
| max | 2088786.00 | 9.30 | 677.00 |

films that concentrate the larger size of tropes. However, the vast majority of films has between 0 and 100 tropes.

As expected, films also follow a non-normal distribution in rating in IMDb website, as it can be seen in Fig 2. Same happens with the number of votes of each film of our dataset in the same web (Fig 3), following a log-normal distribution. There is a lot of films with 0 votes, while there is a little set of about 30 films with more than 1,000,000 votes.

The Table 2, shows the descriptive analysis of the popularity (number of votes) and rating of the films, divided by periods, of the obtained dataset. It must be noted that only IMDb films

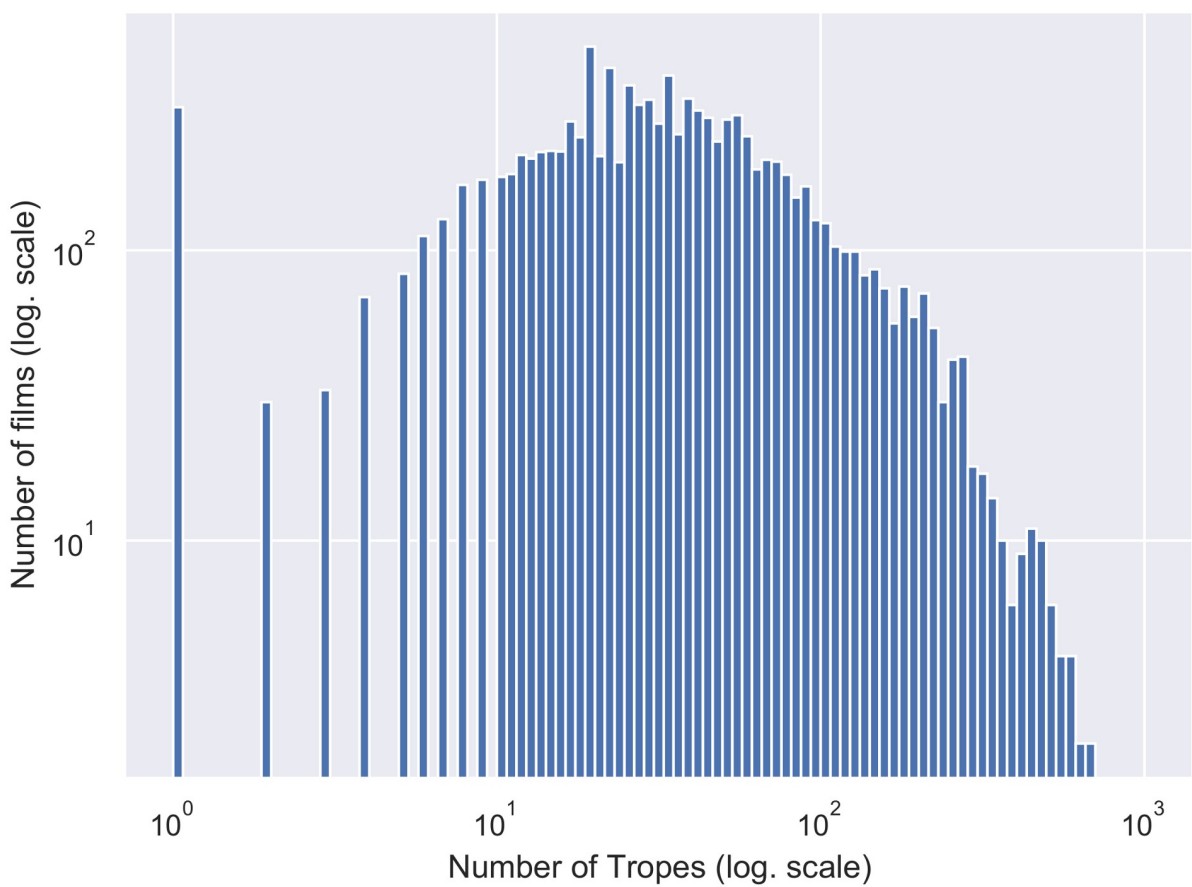

**Fig 1. Distribution of tropes in films (log-log scale applied, zeros were treated adding one to all the values).** Almost all the films have less than 200 tropes. The higher number of tropes in a film is 677.

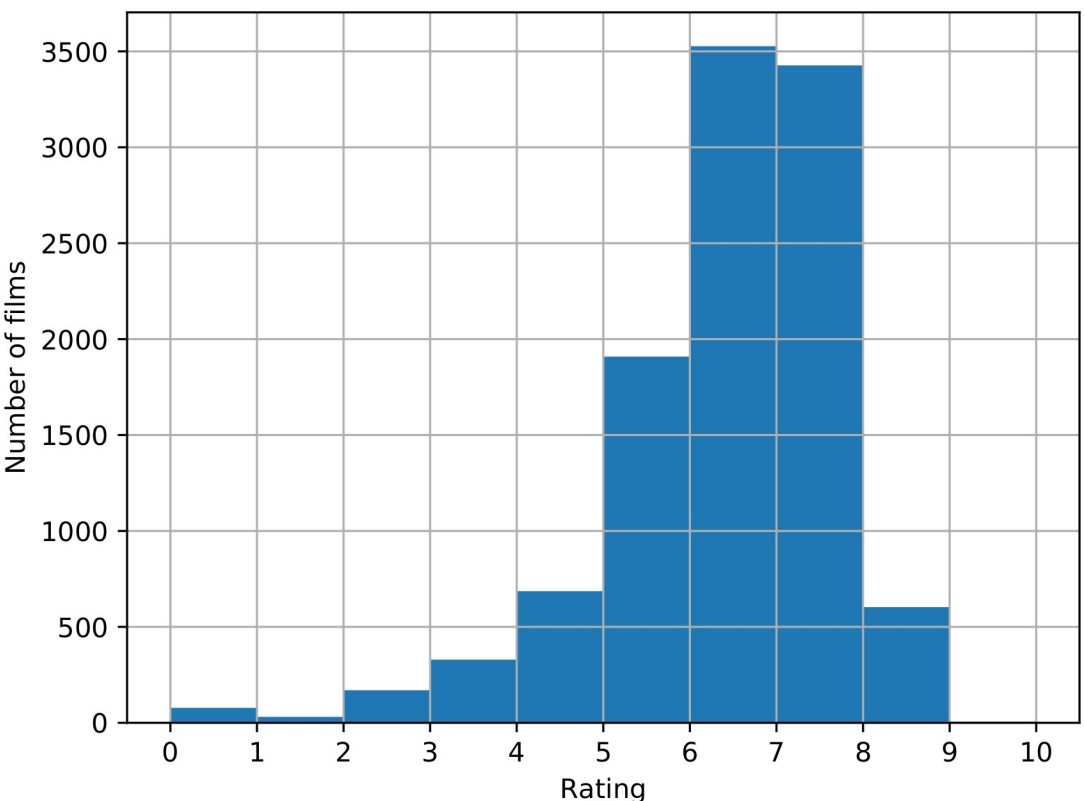

**Fig 2. Rating distribution of all films in the dataset.** Skewness: -1.51. Kurtosis: 3.7.

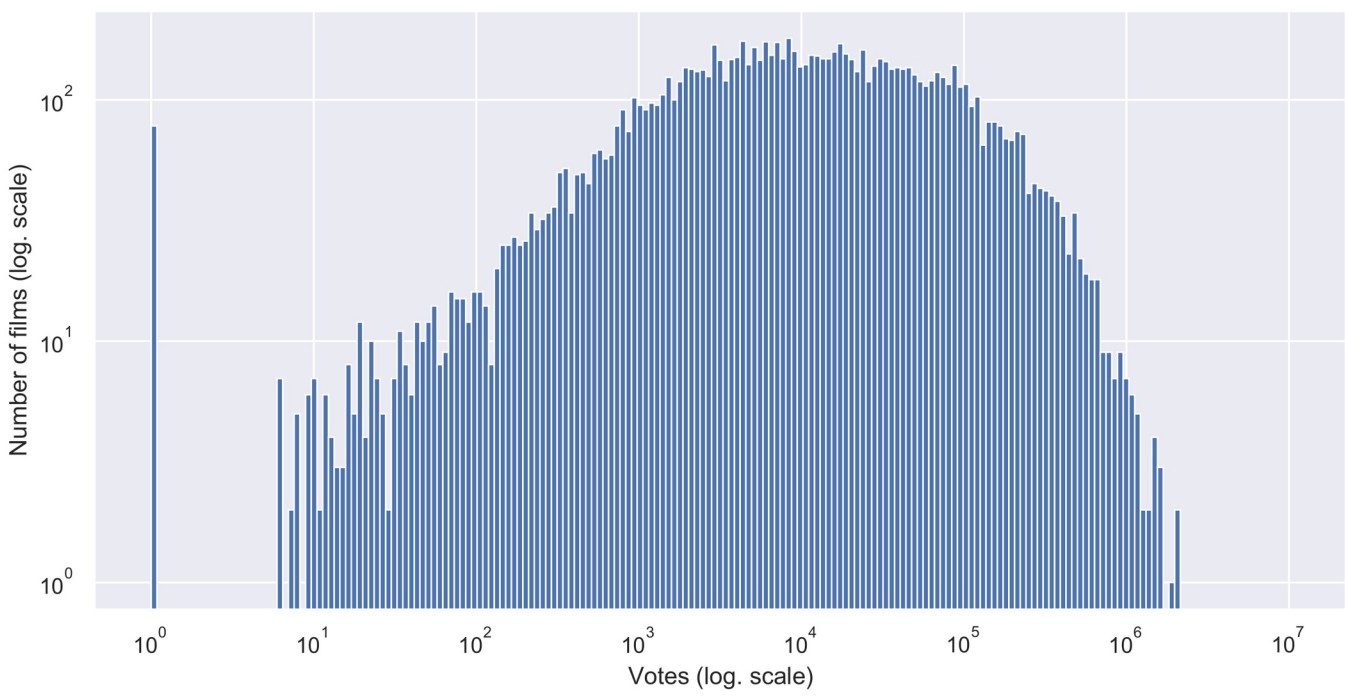

**Fig 3. Distribution of votes (popularity) in films (log-log scale is applied, zeros were treated adding one to all the values).**

**Table 2. Descriptive analysis of votes and ratings by periods for all films of the dataset.** Notice that the data for the period (2000, 2020] is not complete.

| | Votes | | | | | | Rating | | | | | |
|---|---|---|---|---|---|---|---|---|---|---|---|---|
| Periods | count | mean | sth | min | max | median | count | mean | sth | min | max | median |
| (1880, 1940] | 677 | 7335.72 | 24189.94 | 0 | 350943 | 1751 | 677 | 7.02 | 0.85 | 0 | 8.5 | 7.1 |
| (1940, 1960] | 933 | 14323.91 | 42075.84 | 0 | 535004 | 3767 | 933 | 6.84 | 1.34 | 0 | 8.7 | 7.2 |
| (1960, 1980] | 1542 | 21612.29 | 75378.71 | 0 | 1433459 | 4511 | 1542 | 6.58 | 1.35 | 0 | 9.2 | 6.9 |
| (1980, 2000] | 2879 | 49363.80 | 128688.28 | 0 | 2088786 | 12210 | 2879 | 6.24 | 1.28 | 0 | 9.3 | 6.4 |
| (2000, 2020] | 4735 | 71183.88 | 136557.05 | 0 | 2055225 | 20686 | 4735 | 6.17 | 1.35 | 0 | 9.3 | 6.4 |

that have tropes on the TV Tropes website are counted, not the whole IMDb dataset. As it can be seen, the number of films in each period increases over time, with the last 20 years being the period with the largest number, almost half of the dataset. As mentioned on [5], current films receive more attention on the TV Tropes website than old films.

This can also be seen in Fig 4, where is a clear tendency of increasing films analysed in the dataset over the years. In fact, since the 2000s, the amount of films is considerably larger than the rest of the decades.

Moreover, these films have more tropes analyzed in TV Tropes than old films on average. An interesting fact from the IMDb data, also discussed in the paper [5], is that current films tend to receive more votes than older films. Interestingly, and as also shown in Table 2 the average rating for each period decreases over time.

Summarizing, it is clear that there may be some bias in the dataset regarding the number of tropes per film and its popularity, since it has been obtained from two sources based on crowd-sourcing. This is a fundamental difference from applying the methodology in the field of scientific production analysis, where the data to be analysed (citations, keywords) are generally well defined and fixed. In this case, there may be tropes in certain films that are not be included the

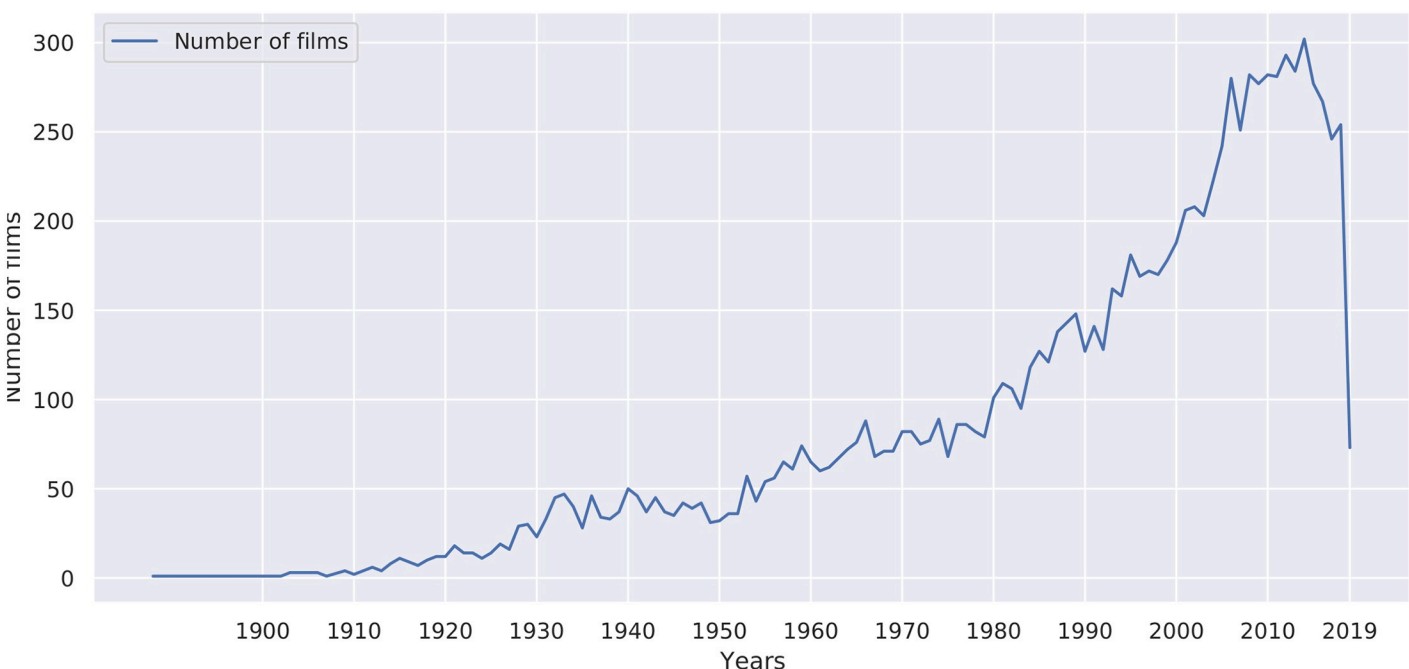

**Fig 4. Number of films by year in the dataset extracted from TV Tropes.** The dataset does not include all films of 2019, as it was generated before the year ended.

**Table 3. Percentage of film genres in the dataset.** NOTE: A film can have more than one genre.

| Genre | % | Genre | % | Genre | % |
|---|---|---|---|---|---|
| Drama | 48.18 | Fantasy | 8.77 | Musical | 2.01 |
| Comedy | 34.5 | Mystery | 8.21 | Western | 1.88 |
| Action | 20.3 | Family | 5.68 | Sport | 1.66 |
| Horror | 17.96 | Biography | 5.3 | Film-Noir | 1.02 |
| Crime | 16.09 | History | 3.5 | Animation | 0.84 |
| Romance | 15.4 | War | 3.19 | None | 0.11 |
| Thriller | 14.66 | Short | 2.94 | Adult | 0.1 |
| Adventure | 14.56 | Music | 2.87 | Game-Show | 0.02 |
| Sci-Fi | 9.54 | Documentary | 2.2 | News | 0.02 |

dataset because users have not analyzed that film in more depth because it is older. Knowing this, it would not be fair to say, for example, that old films were simpler than the current ones because they have fewer tropes. Moreover, this will be important, for example, when conducting a study on emerging/decaying trope communities using strategic diagrams, as we will do later in the paper. Therefore it is important to understand the distribution of data to be used at each moment when applying our methodology. This answers RQ1.

As mentioned in the introduction, film genres are one of the best known and most widely used ways of classifying films. Therefore, it is useful and necessary to obtain how the films are distributed in these genres, so that when we perform some kind of analysis of the tropes (including overlapping analysis) we can better understand the results.

Table 3 shows that almost half of the films in the dataset (48.18%) are Dramas, followed by Comedies and Action films, with 34.5% and 20.3% respectively. Next, there are other minor genres, mostly the ones related to the setting, such as Western, War or Sci-fi, with less than an 8% of presence. There are quite marginal genres, such as Adult, Game-Show and News, with less than 0.1%. The universal genres, mentioned above, are the most common, except for Mystery, which is overtaken by some setting genres, such as Adventure and Sci-Fi.

Understanding this genre distribution is also important for analyzing results or obtaining information. For example, if filmmakers want to find the most interesting tropes to create documentaries it is clear that this genre is not as well represented as the others, possibly because it has less interest for TVTropes users as explained before, and they will have to filter through this genre before making their analysis.

## 4.2 Overlapping analysis

In order to discover the differences among genres based on the tropes (RQ2) we have computed the Jaccard Index for all the genres with all tropes, the 100 most voted tropes and the 100 most rated tropes.

Fig 5c shows the heatmap of the Jaccard index between the tropes belonging to the different genres. As it can be seen, there is a complete overlapping of tropes between marginal genres, such as Short and Documentary, and War and History. Although this association may make sense at first glance, it can also be explained because the number of tropes in those genres is lower and less specific than in other genres. Some genres have no relation with the others (for example, Animation and Film noir, or even Western), that can be explained because they combine a low number of films, but also because they are less general than the others. However, Biography, being a minor genre (5.5% of the films), has a lot of tropes in common with more extended genres, such as Comedy and Crime, but an extremely high overlapping with Drama.

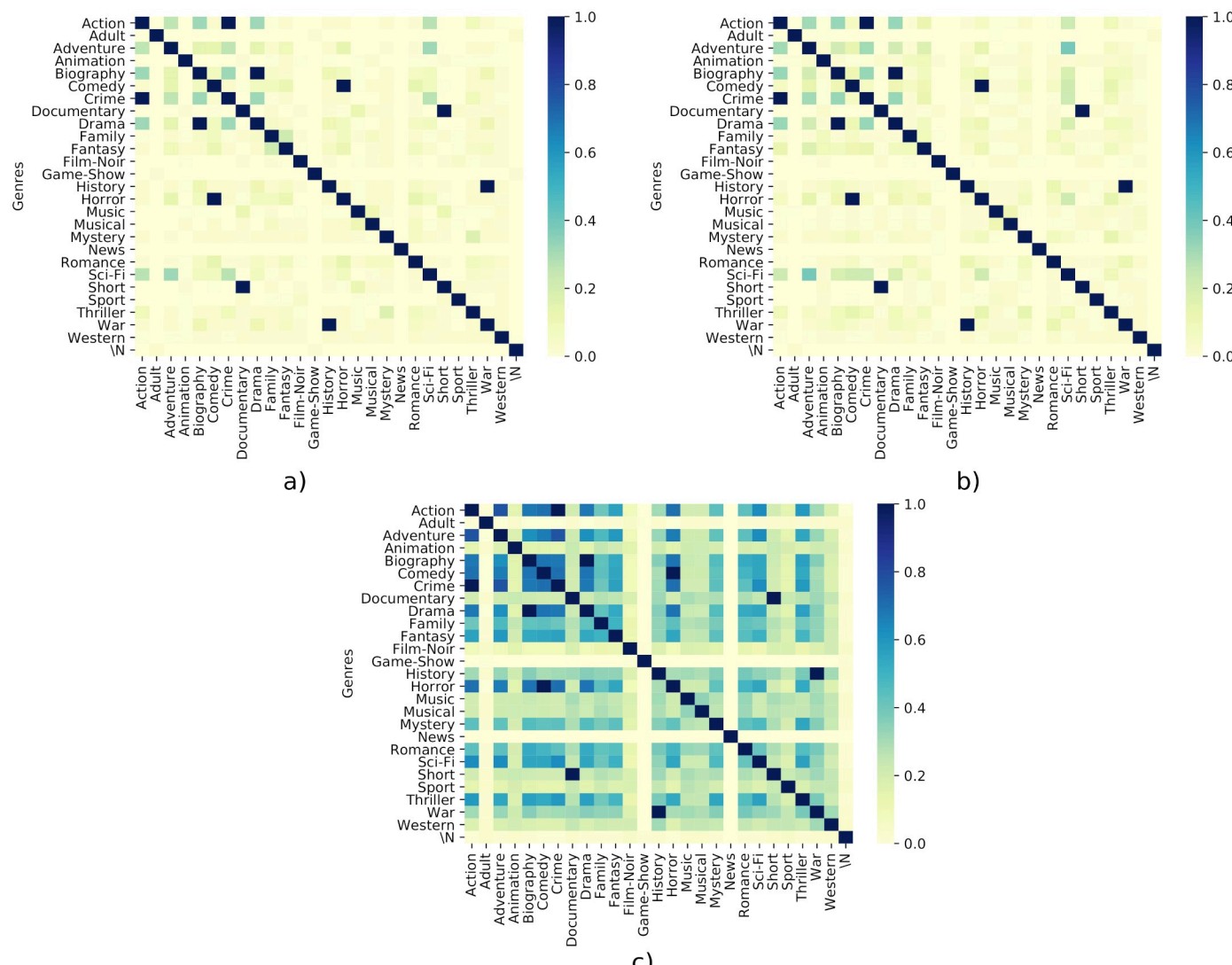

**Fig 5. Jaccard index heatmaps of the overlapping of tropes.** "\N" is used to denote that a film does not belong to a genre. a) Heatmap of the overlapping of tropes with respect to the top 100 rated films. b) Heatmap of the overlapping of tropes with respect to the top 100 voted films. c) Heatmap of the overlapping of tropes with respect to all the tropes of the genres.

The interesting issue of this Fig is that there are more relation between the universal genres (Action, Comedy, Drama, Horror, Mystery, Romance and Thriller), but also other broad genres such as Biography, Crime, Adventure, Fantasy and Sci-Fi. Because we are using all the films of the dataset, it is probably that at least one film of a genre has a trope in common.

Focusing on the top-rated and top voted films, Fig 5a and 5b show some differences concerning the previous overlapping. In this case, while these two new Figures are similar, more overlapping exist between Sci-Fi with Comedy, Adventure and Horror in the most popular ones. However, with respect to the general overlapping (Fig 5c) they show less overlapping between genres with respect to the previous one. Only high overlapping exists between the next pairs: Crime-Action, Horror-Comedy, Short-Documentary, War-History. This means that using only the most popular or top-rated films to study the overlap of genres is not enough

to get an overall picture, as it is clear that viewers prefer little overlap between genres in general.

## 4.3 Co-tropes community detection analysis

So far we have discovered that there is a certain bias in the dataset, both at the level of interest of the films (there is more popularity in the newer films, and they have more detected tropes), and at the level of the genres. The next step in our methodology is to obtain the different trope communities, and to discover if the films belonging to each of these communities also receive more popularity or ratings. Once these communities are found, they can be classified in the different quadrants of a strategic diagram. In this way, filmmakers will not only be able to choose the tropes from those communities related to their current idea, but they will also be able to choose from those communities with higher ratings, or explore the emerging communities (always taking into account the dataset bias). In addition, this step will also offer a view of the troposphere in its current state.

The results of applying the Leiden algorithm to the co-trope network indicate the existence of 42 communities, covering from 10,370 to 1 films, with an average of 2,456.52 ± 2,316.71 films in each. 8 of these communities only have one film.

To confirm that there is a statistical significance relationship between the rating and popularity of the communities, first a Kolmogorov-Smirnov test has been performed to determine if these metrics follow a normal distribution. All p-values obtained in the sets are less than 0.05, so a non-parametric test is necessary. In this case we have used the Kruskal-Wallis test, since the number of samples is independent, of different size, and samples do not follow a normal distribution. The p-value obtained by the Kruskal-Wallis test also obtains a p-value lower than 0.05, both for the rating and for the popularity, indicating that there are significant differences between the sets. Therefore we can confirm our hypothesis that there are differences between the rating and popularity of film communities obtained from tropes (RQ3). However, this does not mean that using specific tropes guarantees better ratings, but rather that, in general, the films that use them have better ratings.

Table 4 describes some of these communities, as well as the metrics of number of films in each community, examples of films, and ratings and average number of votes. These communities can be identified from the central trope, and this is what will be done throughout the paper. The metrics of centrality ($c$) and density ($d$) of each community are also shown in the table, and they will be used later to visualize their degree of development.

As it can be seen, some of these communities are quite recognizable: for example, the tropes in the community identified by the central trope *EldritchAbomination* are related to demonic and supernatural elements, while in others, genres such as Western (*MediumAwareness*) can be detected, in which tropes such as *Gunslinger*, *RidingIntoTheSunset* or the *BountyHunter* appear. Other communities show odder or uneasier themes (*EarlyInstallmentWeirdness*), or related to technology (*LaserGuidedAmnesia*) or to disasters (*BigReddButton*). The first community (*ShoutOut*) is particularly interesting, since instead of relating to a specific theme, it groups the best-known tropes, such as *BigBad* or *ChekovsGun*, and elements inherent to the narrative, such as sad endings. In this community, there are 10,370 films, practically the size of the dataset.

Because of the large number of films in each community, examples may not be as obvious. For example, in the aforementioned *EldritchAbomination* community, light comedies such as *Jack And Jill* appear. This also can be explained because, as shown previously when describing Fig 5 the Jaccard index between their tropes was very high, therefore they may appear together in the community.

**Table 4. Performance measures from 10 arbitrary communities obtained with the Leiden algorithm for the co-tropes network.**

| Id | #Films | Central Trope | Tropes | Films | Density | Centrality | Mean/Std. Rating | Mean/Std. Votes |
|---|---|---|---|---|---|---|---|---|
| 1 | 10370 | ShoutOut | ShoutOut, BigBad, ChekhovsGun, Foreshadowing, OhCrap, BittersweetEnding, TitleDrop, DeadpanSnarker, DownerEnding, Jerkass | The Legend of Bagger Vance, A Tale of Two Sisters, Blow-Up, House on Haunted Hill, Stand Up Guys, Colossal, Cry-Baby, Original Sin, The Astronauts Wife, Thirteen Days | 183,72 | 186326,98 | 6.37 ± 1.33 | 50520.03 ± 119953.41 |
| 2 | 5013 | EldritchAbomination | DemonicPossession, EldritchAbomination, OurVampiresAreDifferent, DoesNotLikeShoes, SealedEvilInACan, Satan, PsychicPowers, HumanSacrifice, HollywoodAtheist, OurGhostsAreDifferent | The Invasion, The Mothman Prophecies, Conan the Destroyer, Frailty, Jack and Jill, The Sandlot, Swingers, Burlesque, The Uninvited, The Florida Project | 167,64 | 73999,37 | 6.31 ± 1.36 | 71130.22 ± 146138.64 |
| 3 | 4850 | TwentyMinutesIntoTheFuture | TwentyMinutesIntoTheFuture, ItCanThink, FeetFirstIntroduction, AlienInvasion, MegaCorp, TheRival, ScreamingWoman, GreenAesop, UncannyValley, HumansAreTheRealMonsters | xXx: Return of Xander Cage, Geostorm, Heathers, Us, Flubber, Hardcore Henry, Universal Soldier, The Smurfs, Beasts of the Southern Wild, Scrooged | 207,37 | 85646,63 | 6.34 ± 1.36 | 77580.52 ± 157457.59 |
| 4 | 3678 | DramaticUnmask | DramaticUnmask, FaceFramedInShadow, BankRobbery, BadLiar, NoHonorAmongThieves, EvilWearsBlack, BluntYes, IWarnedYou, CreateYourOwnVillain, RayOfHopeEnding | Crimson Tide, Sicario: Day of the Soldado, I Saw the Devil, Faster, Sphere, Attack the Block, Searching, Crazy Rich Asians, The Ladykillers, Saw VI | 229,98 | 54172,12 | 6.51 ± 1.31 | 93440.54 ± 171231.86 |
| 5 | 3229 | ComicBookAdaptation | YouCantFightFate, TimeTravel, ImplausibleDeniability, ComicBookAdaptation, SetRightWhatOnceWentWrong, RedScare, StableTimeLoop, SpoilerTitle, ThrowTheDogABone, OmnicidalManiac | T2 Trainspotting, Caddyshack, My Cousin Vinny, Safe, Julie & Julia, Bridget Jones: The Edge of Reason, The Forbidden Kingdom, They Live, Serpico, Yojimbo | 236,5 | 46619,06 | 6.49 ± 1.3 | 96248.88 ± 177620.28 |
| 6 | 3162 | MediumAwareness | MediumAwareness, TheGunslinger, NoFourthWall, TheSheriff, BestialityIsDepraved, BountyHunter, CompanionCube, RidingIntoTheSunset, YouNoTakeCandle, GoodShepherd | Hunt for the Wilderpeople, Runaway Bride, Cheaper by the Dozen, The American, West Side Story, National Lampoons Vacation, Colombiana, Sahara, eXistenZ, Alita: Battle Angel | 174,2 | 28804,15 | 6.55 ± 1.27 | 86994.61 ± 170764.69 |

(*Continued*)

**Table 4.** (Continued)

| Id | #Films | Central Trope | Tropes | Films | Density | Centrality | Mean/Std. Rating | Mean/Std. Votes |
|---|---|---|---|---|---|---|---|---|
| 7 | 1996 | BigRedButton | JiveTurkey, AttackOfTheKillerWhatever, ReptilesAreAbhorrent, BigRedButton, PlayingAgainstType, TheCassandra, Bizarrchitecture, WhiteMaskOfDoom, ActionHero, MadScientistLaboratory | Legion, Touch of Evil, Willow, Hot Shots!, Sixteen Candles, Power Rangers, Dan in Real Life, Deliverance, Bedazzled, Death Becomes Her | 199,64 | 23291,64 | 6.27 ± 1.37 | 92326.69 ± 172328.62 |
| 8 | 1547 | ItWasHisSled | ExtraExtraReadAllAboutIt, YourMindMakesItReal, TheCanKickedHim, ItWasHisSled, HostageForMacGuffin, GunStruggle, TheOphelia, DreamWithinADream, DeathIsCheap, CreepyTwins | Across the Universe, Catwoman, Push, Upgrade, Entrapment, Carol, Bicentennial Man, Lights Out, End of Days, Precious | 162,28 | 18107,11 | 6.51 ± 1.28 | 101116.32 ± 186806.43 |
| 9 | 4085 | EarlyInstallmentWeirdness | TheVamp, CrazyJealousGuy, ParentalIncest, FrameUp, MaleFrontalNudity, OnlyOneName, ThePlace, BuryYourGays, SinisterMinister, EarlyInstallmentWeirdness | Year One, Licence to Kill, The Devils Rejects, Small Soldiers, The Legend of Zorro, The Last Witch Hunter, Halloween, Money Monster, Another Earth, The Last Boy Scout | 268,24 | 18835,36 | 6.48 ± 1.26 | 85916.57 ± 167876.93 |
| 10 | 851 | LaserGuidedAmnesia | LaserGuidedAmnesia, MalignedMixedMarriage, SeparatedByACommonLanguage, UnfinishedBusiness, LiteralGenie, CulturalPosturing, YourCostumeNeedsWork, StabTheSalad, TheGreys, Invisibility | Priest, The 6th Day, The Disaster Artist, Robin Hood: Men in Tights, Casper, The Dark Tower, Big Trouble in Little China, Under the Skin, The Time Machine, Star Trek II: The Wrath of Khan | 171,79 | 10808,2 | 6.44 ± 1.3 | 108922.87 ± 175284.13 |

Once we have seen that there are differences between communities, screenwriters can use combinations of tropes that appear in the highest rated or most popular communities to inspire them to develop new scripts, although as stated above, this does not automatically guarantee success. However, it may also be interesting to seek inspiration from those communities that are still developing and avoid those that are in decline. In addition, if the type of film to be developed is very specific, inspiration should also be sought from communities related to that type of film. That is why we will use a Strategic Diagram to visualize the degree of development of these communities and to be able to choose among the most appropriate ones.

The strategic diagram obtained is shown in Fig 6. In the upper-right quadrant are the *motor* (central and developed) communities. It has high *c* and high *d*. The communities in this quadrant have nodes that are strongly connected to each other, but also serve as support for the rest of the communities. In scientometrics it is considered the file core and their position is strategical. The works in this quadrant are treated systematically by a well-defined group of

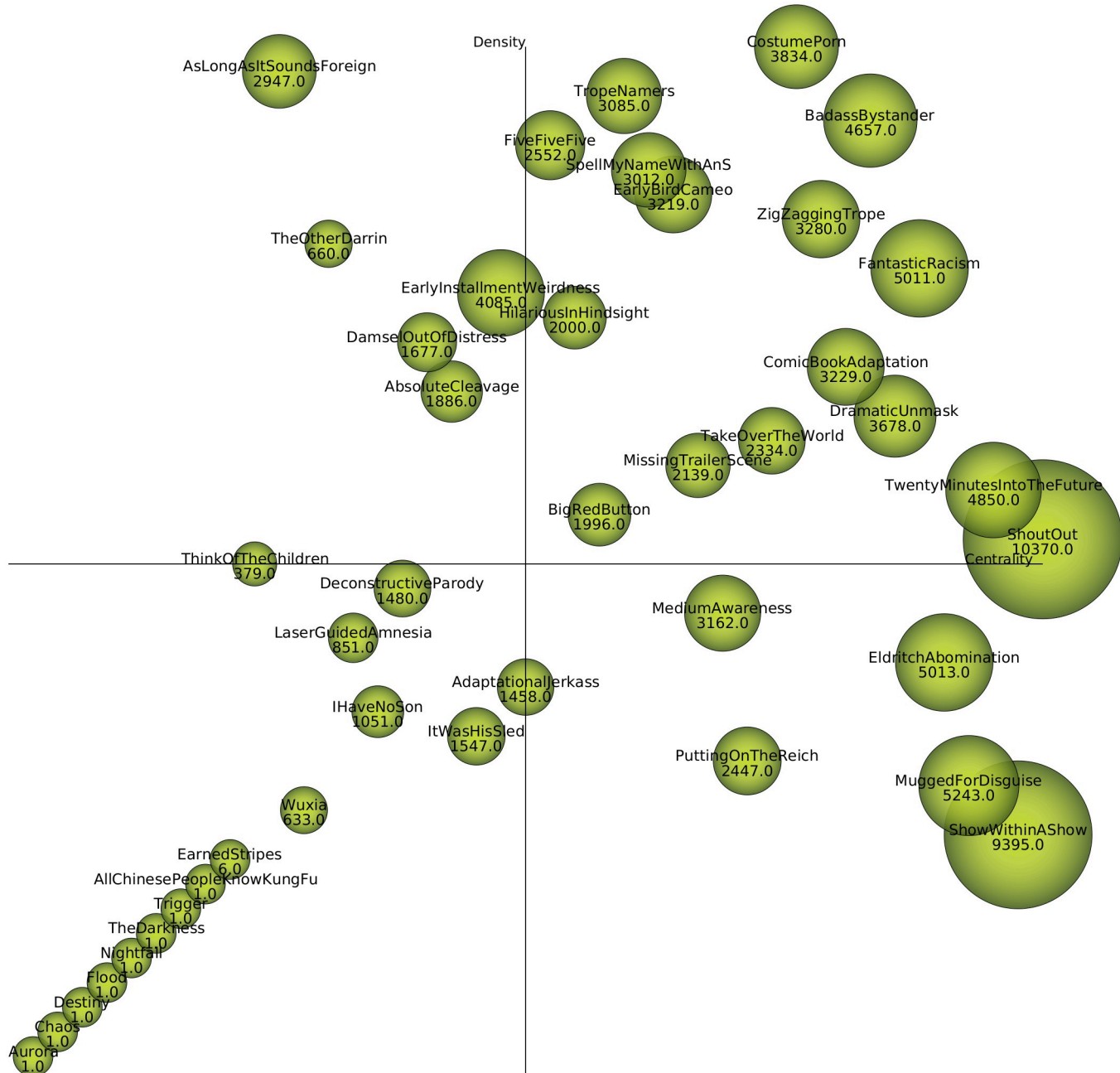

**Fig 6. Strategic diagram of the communities obtained from the co-tropes network.** Depending on centrality (x-axis) and density (y-axis) of each community, they can be placed in any of the 4 quadrants whose center is in (0.5,0.5). From left to right and top to bottom: specialized/peripheral, motor (central/developed), emerging/decaying and transversal. Size of the dots is proportional the number of movies in each community (also shown in numeric value).

researchers over a long period of time. In the case of movies, we can say that the tropes are very related to each other, but they are used as a base in many different types of films, also over a long span of time. Therefore they may be very known for the general public. That is the reason the vast majority of the communities detected (16 out of 34 with more than one movie) are in this quadrant (including most of the described in Table 4.

In the lower-right quadrant are the *transversal*, or basic/general communities (low *d* and high *c*). In scientometrics these are themes that are not developed, but which serve to develop the rest of the research in the area. For example, if we are analysing works related to cancer research, neural networks are not developed in this scientific field, but they are used in a lot of works. In the case of this paper, we can consider sets of tropes that are in many films, but that are not usually used all at once. That is, they are becoming mature, but are not yet the object of investments in other movies. A few of the communities found are in this quadrant, *EldritchAbomination* being one of them (this community included comedies and horror films). The *ShowWithinAShow* community includes many tropes common to comedies (mostly romantic), especially tropes related to rival female characters. However, it also serves to characterize films like *Darkman*.

In the upper-left quadrant (high *d*, but low *c*) are the *specialized and peripheral themes*: highly developed, but isolated, themes are considered. If we apply this explanation to case of the tropes, we can consider they are very cohesive with each other, but they are not so related to other trope communities. That is, when these tropes are used, they are only used together and not with other tropes, so that imply they only appear in highly specific films, or because have been marginalized, generating less and less interest. In fact, tropes that appear in *EarlyInstallmentWeirdness* can be considered controversial (*ParentalIncest*, *BuryYourGays*).

Finally, in the lower-left quadrant (low *c* and low *d*) are the *emerging or decaying* themes. There are no communities that are supporting them, either because they are very marginal themes or because they are no longer important, or on the contrary, they are very new. In this case, an expert analysis is needed to study these themes. That is the reason why the 8 communities with only 1 movie appear here. Also, communities with more movies, such as *LaserGuidedAmnesia*, although thematically similar to *BigRedButton* (that appears in the general quadrant), are represented here. Fig 7 shows the distribution of the movies of these communities using a heatmap. Results shows how all the movies in the communities of this quadrant are inclined towards recent years (but as explained before, the whole dataset is unbalanced to more recent movies). Therefore we cannot assure there exist declining communities. However, some of the communities are more distributed along extended periods of time, while others are condensed in more recent years. In addition, there has been an apparent increase in the number of films from some communities in recent years (*ItWasHisSled*, *DeconstructiveParody*, *AdaptionalJerkass*), indicating that there is more interest in the tropes represented in those communities.

After analysing the position of the tropes communities have been able to get an overview of which of these tropes may be interesting to develop in the future, which are a safe bet, or which of them may produce less interest. This answers RQ4, showing that the different communities of tropes have different degree of development.

As the next step of the methodology, a co-trope graph has been generated. Fig 8 shows the spanning tree of the co-trope network, obtained in the same way as the co-word networks are obtained in the bibliometric analysis. In these networks, each word (trope) is a vertex, whose edge to another trope measures the number of films that both share. The spanning tree facilitates the visualization of these nets. However, since we have to represent 25,776 tropes, we have had to limit the visualization. To do this, we have filtered the network keeping only the following: the nodes with a frequency higher than 30, the edges with a weight higher than 60 and the nodes which have at least one edge. As it can be seen, the most important tropes are *OhCrap* (something really bad is about to happen), *Shoutout* (references to other works), *Foreshadowing* (a clue that predicts an event) and *BigBad* (the villain in general, this trope includes many types of evil or antagonist characters). From these tropes, linked together, hang related tropes in a star-shaped way.

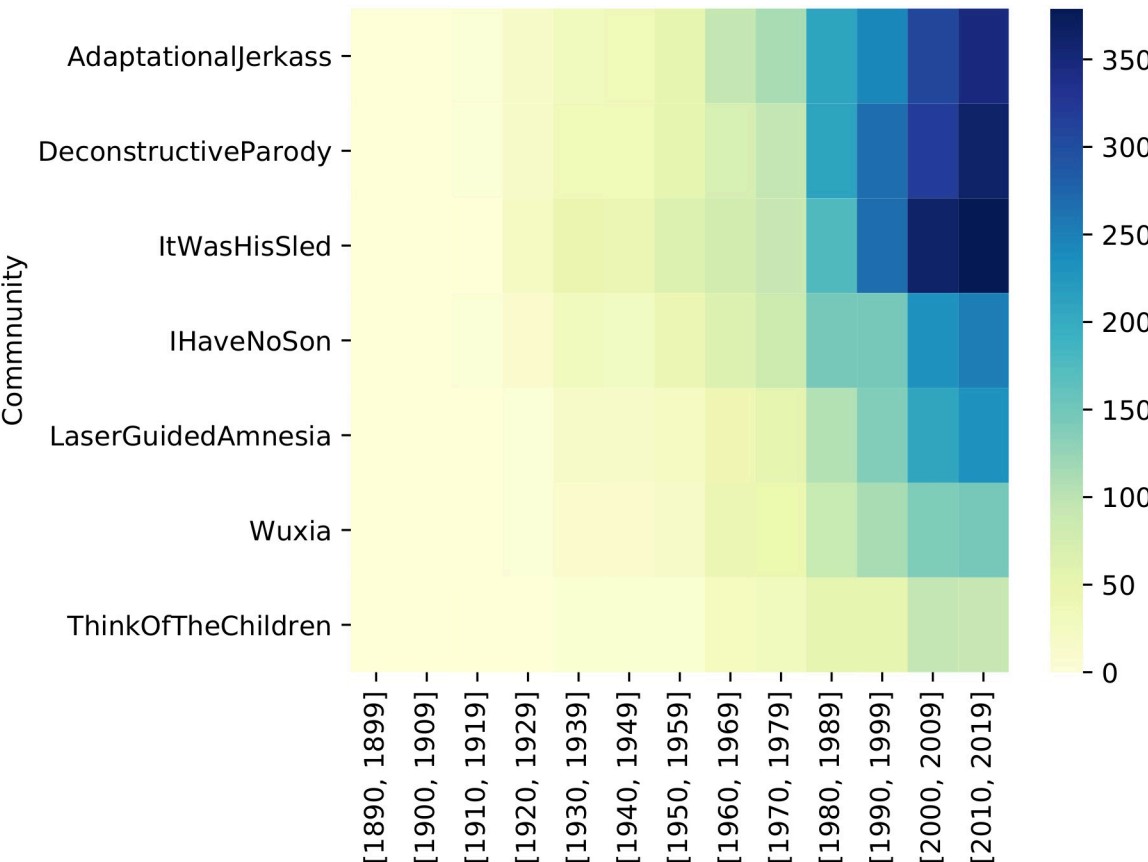

**Fig 7. Distribution by decade of the movies of all communities detected in the emerging/dissapearing quadrant of the strategic diagram.**

Of all these main tropes, *ShoutOut* is the only one that is not directly related to the plot, but it is the one that is most repeated in the dataset. This is why it is related to other tropes not related to the plot, such as *TitleDrop* (mention of the film title in the dialogue), *FanService* (references only understood by fans), or *RunningGag*.

*BigBad* is related to tropes that have to do with serial killers (*AxCrazy*), *AntiHeroes*, or Big-Bad's right hand: *TheDragon*. One of the branches that emerge from *BigBad* is related to groups of characters that usually face evil together: *TheChick*, *TheLeader*, *FiveManBand*, *TheSmartGuy* and *TheBigGuy*.

*OhCrap* is close to tropes that indicate that something bad is going to happen, such as *TooDumbToLive* (the character does not do what the viewer would do in real life and dies) or *KickTheDog* (the villain does a cruel and unnecessary action).

Finally, *Foreshadowing* indicates relationship with tropes that affect the plot (*TheReveal,BittersweetEnding*, *HeroicSacrifice*). This is a technique that is present, and besides several times, in many movies and TV series; in these generally previous episodes foreshadow what is going to happen further down the season. In a way, it is similar to *Chekhovsgun*, although this one usually refers to a specific device that will *foreshadow* its use later on in the story.

The genre distribution of the communities found can be seen in Table 5. Although there is much similarity in the distribution, it can be seen how some of these communities are more focused on specific genres. For example, *BigRedButton* is more related to the genre of Horror and Science Fiction than the other communities. Therefore, depending on the gender of the

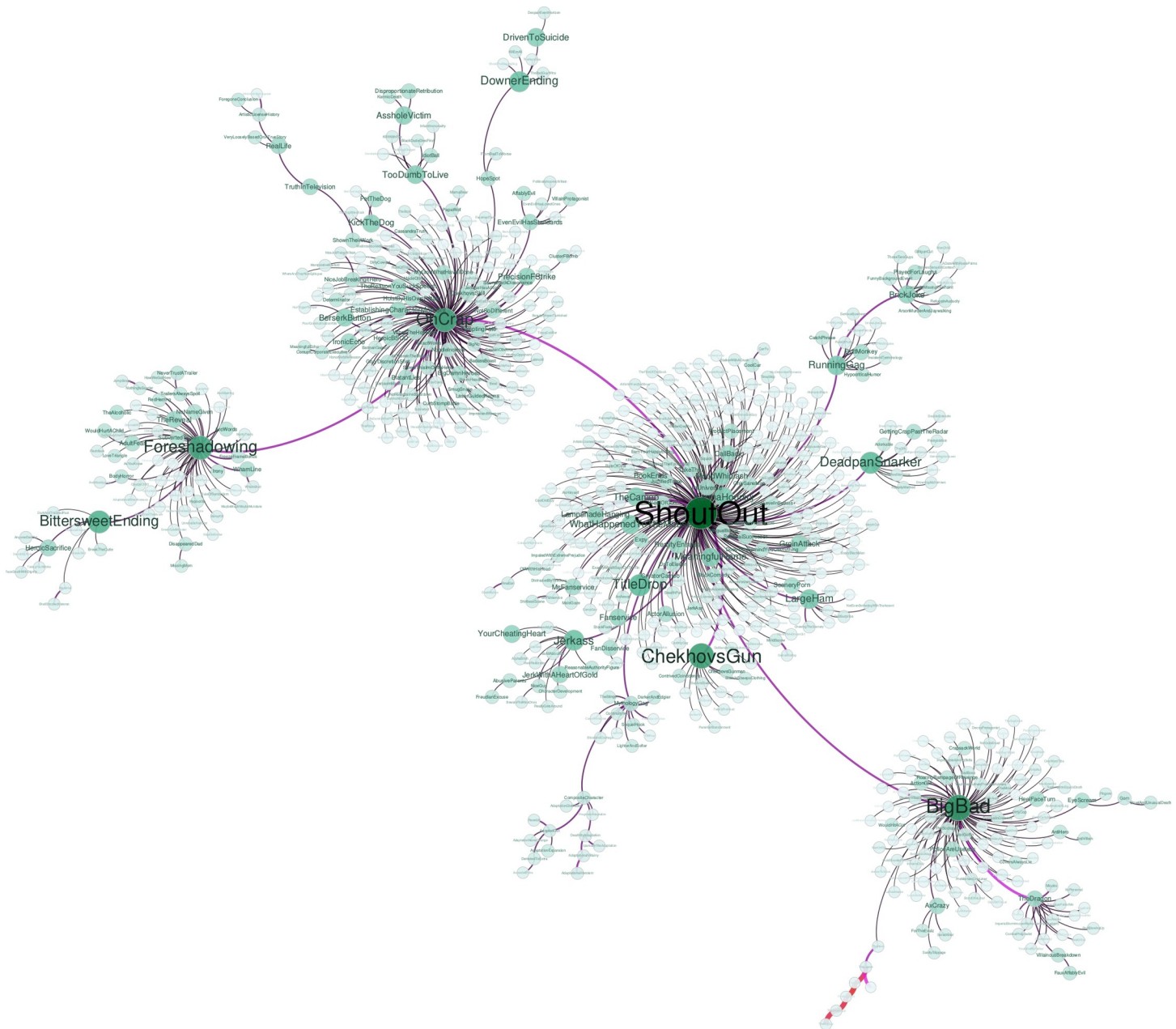

**Fig 8. Co-tropes spanning tree of the co-occurrence network.** The larger the node more films have the trope. The larger the edges, more films both tropes have in common.

story to be developed we can restrict those communities that are more appropriate to our purposes.

## 4.4 Co-films community detection analysis

In this case, and after applying the Leiden algorithm, a smaller number of communities have been detected, only 10, of which five are composed by a single film. As in the previous subsection, the same methodology has been applied to confirm the statistical significance among the groups obtained. As these are also samples that do not follow a normal distribution, the

**Table 5. Rate of genres for 10 arbitrary communities obtained with the algorithm.**

| Id | CentralTrope | Action | Adventure | Comedy | Crime | Drama | Fantasy | Horror | Romance | Sci-Fi | Thriller |
|---|---|---|---|---|---|---|---|---|---|---|---|
| 1 | ShoutOut | 20.37 | 14.71 | 34.5 | 16.16 | 48.09 | 8.84 | 18.21 | 15.26 | 9.63 | 14.83 |
| 2 | EldritchAbomination | 23.72 | 19.19 | 32.68 | 12.27 | 40.45 | 14.58 | 26.49 | 11.65 | 13.76 | 15.16 |
| 3 | TwentyMinutesIntoTheFuture | 28.91 | 21.71 | 29.86 | 14.97 | 41.18 | 11.24 | 22.39 | 9.86 | 17.22 | 16.87 |
| 4 | DramaticUnmask | 31.16 | 21.34 | 33.5 | 18.76 | 43.47 | 10.14 | 15.39 | 11.28 | 12.59 | 16.23 |
| 5 | ComicBookAdaptation | 29.64 | 23.32 | 32.64 | 14.96 | 42.46 | 11.46 | 16.38 | 10.37 | 15.52 | 15.83 |
| 6 | MediumAwareness | 27.13 | 21.44 | 38.14 | 17.05 | 42.09 | 10.34 | 13.06 | 12.78 | 11.7 | 13.85 |
| 7 | BigRedButton | 30.06 | 23.65 | 29.81 | 11.27 | 32.16 | 13.48 | 29.41 | 8.52 | 20.74 | 17.89 |
| 8 | ItWasHisSled | 28.57 | 22.24 | 29.67 | 16.87 | 41.89 | 13.32 | 21.72 | 10.41 | 14.09 | 17.0 |
| 9 | EarlyInstallmentWeirdness | 26.51 | 17.92 | 31.33 | 20.39 | 45.51 | 9.99 | 19.68 | 11.95 | 10.77 | 17.94 |
| 10 | LaserGuidedAmnesia | 29.96 | 23.62 | 34.55 | 12.69 | 36.43 | 14.92 | 24.09 | 9.99 | 19.15 | 16.8 |

Kruskal-Wallis test for rating and popularity also gives a p-value lower than 0.05, confirming that there are significant differences in all the measures between the different groups.

Focusing on the 5 communities with more than one film, interesting differences can be seen, as described in Table 6. Moreover, they show much more difference between the genres, as shown in Table 7, where each community stands out in some specific genre.

The community with the most tropes (and the most votes) has *"Thor"* as its central film, and it is composed of films related mainly to superheroes. The most abundant genres are Adventure, Action, Comedy and Fantasy. The tropes of this community are related to action characters (*ArmsDealer*, *TheLastOfHisKind*), or dangerous environments (*IceBreaker*). Others are related to humor: *TheOneThingIDontHateAboutYou* (the villain has a funny quality), *ThatCameOutWrong* (an innocent word game that can be confused with something dirty because the character has not stopped to think about it). There are also tropes related to non-verbal communication of characters (*IDoNotSpeakNonverbal*, *Narrative-Shapeshifting*), very common in this kind of films (for example, aliens like Groot from *"Guardians of the Galaxy"*).

The rest of the communities have a similar number of tropes, between 8000 and 14000. Community 0 is formed by comedies, many of them romantic. This is also reflected in the high percentage of both genres with respect to the rest of the communities. It has tropes based on love or complex relationships, such as *RelationshipUpgrade* (the moment when two characters become an official couple) or *ButNotTooBi* (a bisexual character only relates to one sex), or *IdentityAmnesia* (the character has forgotten who he is). Other tropes related to humor: *Non-FatalExplosions* (explosions that are not lethal) or *Exit-PursuedByABear* (the villain runs away while being chased by an animal).

Community 2 is composed mostly of Thrillers and horror films, with tropes largely related to the subject matter. There is a group of tropes that define the evil character (*EvilSoundsDeep*, *VocalDissonance*). Another very common one is *DwindlingParty*: the members of the group are dying one by one.

*"Mullholland Drive"* is the centerpiece film of community 3. Despite having a percentage of genres very similar to community 0, the films of this community also include more serious films, such as *"Birdman"* or *"Donnie Darko"*. However, comedies with a certain degree of fantasy and adventure (*"Hot Tub Time Machine"*, *"The Cannonball Run"*) appear here. That is why there are some tropes related to dark films (*FacialHorror*, *CosmicHorrorStory*) or comical (*ZeroChops*: the characters fight without knowing how to fight in a funny way).

Finally, community 4 includes the highest percentage of Drama, including war films (*"The Thin Red Line"*), family terror (*"The Babadook"*), or dark science fiction (*"Children of men"*, *"The Butterfly Effect"*). This is the community with the highest average score. The tropes that

**Table 6. Co-films performance measures.**

| Id | #Tropes | CentralFilm | Tropes | Films | Density | Centrality | Mean/ Std. Rating | Mean/ Std. Votes |
|---|---|---|---|---|---|---|---|---|
| 0 | 14158 | Bridget Jones: The Edge of Reason | ShowdownAtHighNoon, WhatsAnXLikeYouDoingInAYLikeThis, ButNotTooBi, OldMoney, ExitPursuedByABear, RelationshipUpgrade, EvilTastesGood, WorkingThroughTheCold, NonFatalExplosions, IdentityAmnesia | Bridget Jones: The Edge of Reason, The Baby-Sitters Club, High School Musical, Carry On, Love Actually, The Twilight Samurai, Crazy Rich Asians, Fighting with My Family, Enter the Void, The Favourite | 134.52 | 134803.93 | 6.63 ±0.6 | 122891.23 ± 88600.93 |
| 1 | 22540 | Thor | ArmsDealer, TheLastOfHisKind, ThatCameOutWrong, AnAsskickingChristmas, IceBreaker, NarrativeShapeshifting, PrettyFreeloader, IDoNotSpeakNonverbal, HollywoodLaw, TheOneThingIDontHateAboutYou | Thor: Ragnarok, Thor, X-Men: Days of Future Past, X-Men: Apocalypse, The Dark Knight Rises, X-Men: First Class, TRON: Legacy, Avengers: Age of Ultron, Captain America: The First Avenger, Captain America: The Winter Soldier | 342.54 | 172759.61 | 6.64 ±0.73 | 164811.39 ± 150950.33 |
| 2 | 12894 | Paranormal Activity | DarkReprise, AlliterativeName, VocalDissonance, VillainWithGoodPublicity, VirginTension, DwindlingParty, Sting, EvilSoundsDeep, HostileWeather, SympatheticVillainProtagonist | The Thing, Paranormal Activity, Psycho, The Ring, Se7en, Ju-on: The Grudge, Alien: Covenant, Night of the Living Dead, Laid to Rest, The Grudge | 180.51 | 128023.74 | 6.52 ±0.61 | 131828.44 ± 94028.68 |
| 3 | 10951 | Mulholland Drive | ShrinesAndTemples, HauntedTechnology, FacialHorror, LongLostRelative, ZeroChops, AdaptationalNationality, HarmfulToMinors, DiesWideOpen, LongHairIsFeminine, CosmicHorrorStory | Mulholland Drive, The Parent Trap, LEclisse, The Great Beauty, Donnie Darko, Birdman, From Justin to Kelly, Hot Tub Time Machine 2, Hot Tub Time Machine, The Cannonball Run | 154.61 | 84872.81 | 6.59 ±0.57 | 127683.12 ± 83410.89 |
| 4 | 7980 | Vampire Academy | HisNameIs, ShaggyDogStory, NightmareFace, HistoricalVillainUpgrade, MouthOfSauron, TruthInTelevision, KilledOffScreen, BaldOfEvil, ShootTheFuelTank, RaceAgainstTheClock | Vampire Academy, Children of Men, Wuthering Heights, The Thin Red Line, The Breakfast Club, The Babadook, The Butterfly Effect, Amadeus, Blue Is the Warmest Color, Z-O-M-B-I-E-S | 112.7 | 54620.0 | 6.67 ±0.5 | 136932.54 ± 82832.22 |

**Table 7. Co-films rate of genres.**

| Id | CentralFilm | Action | Adventure | Comedy | Crime | Drama | Fantasy | Horror | Romance | Sci-Fi | Thriller |
|---|---|---|---|---|---|---|---|---|---|---|---|
| 0 | Bridget Jones: The Edge of Reason | 7.44 | 7.76 | 41.83 | 12.53 | 60.87 | 6.15 | 5.03 | 26.7 | 3.74 | 7.92 |
| 1 | Thor | 46.04 | 30.9 | 39.38 | 20.66 | 33.85 | 12.23 | 8.43 | 6.49 | 14.8 | 15.98 |
| 2 | Paranormal Activity | 19.64 | 10.82 | 16.72 | 17.77 | 34.32 | 8.99 | 52.61 | 3.52 | 14.07 | 26.15 |
| 3 | Mulholland Drive | 12.39 | 12.09 | 52.8 | 13.96 | 45.53 | 8.95 | 13.18 | 17.31 | 10.23 | 11.41 |
| 4 | Vampire Academy | 14.2 | 10.12 | 16.62 | 18.73 | 78.85 | 8.61 | 11.18 | 18.88 | 7.25 | 14.05 |

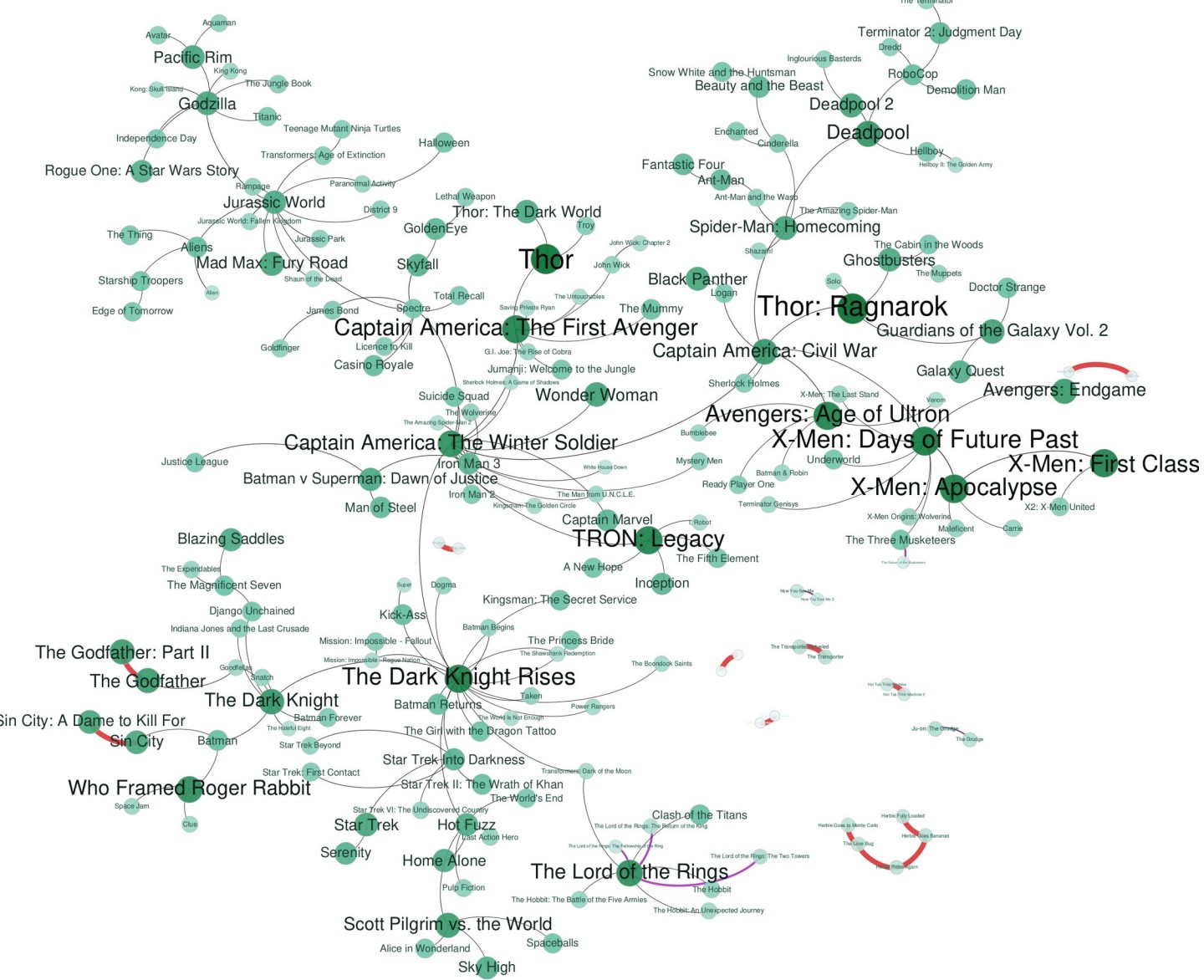

**Fig 9. Co-films spanning tree of the co-occurrence network.** The larger the node more tropes appear in the film. The larger the edges, more tropes both films have in common.

appear in this community represent descriptions of villains: *NightmareFace* (a horrible face), *HistoricalVillainUpgrade* (the historical villain in the film is worse than the real one), *MouthOfSauron* (the villain is never seen directly), *BaldOfEvil* (the villain is bald). Other tropes, like *RaceAgainstTheClock* or *ShootTheFuelTank* are related to the intense rhythm of the film.

As in the previous subsection, we have created a spanning tree that used the films as nodes, and the number of common tropes for the vertices. Also, to facilitate the visualization, we have filtered the network with the same values as before. Fig 9 shows this tree. As it can be seen, the films shown are usually quite commercial, especially superhero films, and specifically the Marvel Cinematic Universe (MCU), linked together. Films from different sagas (James Bond

collection, The Lord of The Rings, Star Trek), or similar themes (aliens, giant monsters, Disney princesses) are also shown together. For example *Robocop*, *Terminator*, *Dredd* and *Demolition man* have in common "future element" and "gun violence", while *The Godfather*, *Goodfellas* and *Snatch* are clearly mafia-themed films. There are a few films disjointed from the rest of the films in the tree, mostly first and second parts, being an interesting mention the *Herbie* saga, indicating that the movies in this saga share very specific tropes between them. The coherence of this graph is also be supported by the fact that films within the same saga that keep the same story are directly connected, to cite some of them, "Jurassic world" (linked to "Jurassic World: Fallen Kingdom" and to "Jurassic Park"), "Thor" (linked to "Thor: The Dark World") or "The Godfather" (linked to its second part) among many others. However, sagas that are known to change actors, path or even genres are also noticeable, for example, "Thor: Ragnarok" is closer to "Guardians of the Galaxy" than to its prequels and "Batman & Robin" is closer to "Avengers: Age of Ultron" than to the other films of the saga.

## 5 Conclusions

The study of tropes in the field of cinema is a very interesting topic for researchers in social sciences and other fields, for example to understand how humans interact and consume this cultural media, or to find inspiration to develop new works. This is the main motivation for our paper, which, however, uses in most cases a generic approach to data analysis that could be leveraged in cultural or media studies. In this paper, we have proposed to use techniques based on scientometrics and complex network analysis to extract information from the tropes that compose the films: specifically to find out if there are any more popular or rated movies based on communities of tropes, and and what is the level of development of these communities. We have used the data available in TV Tropes and IMDb, two collaborative sites edited by enthusiasts on the issue, to generate a dataset formed by 10,766 movies and its associated set of tropes (25,776 different tropes in total). By obtaining some insight on the relationship between patterns (be they cultural, or narrative, since tropes can be any of them) and popularity, we try to find out what are the causes of that popularity and provide a data-driven approach to story, plot or simply pitch generation.

In order to gather these insights, we have initially carried out a descriptive study of the dataset, as well as the study of the overlapping between genres. This study shows a bias in the dataset towards more recent films in the dataset: these films have more user-defined votes and tropes. In addition, there is a large overlap of tropes between some specific genres, such as Action-Crime, History-War or Biography-Drama. These biases should be taken into account in either sense: either to use these tropes, since they are related to popularity, or to try and find out what made old or not so popular movies a success in some sense outside the (crowdsourced) troposphere.

Regarding the analysis of the communities obtained from the co-trope network, a number of recognizable communities have been obtained, showing significant differences in rating and popularity, although the genre distribution of these communities is similar. Also, these communities show a different degree of development. As indicated in the results, there are communities in the emerging quadrant that are precisely related to previous adaptations or versions (ItWasHisSled, AdaptationalJerkass and DeconstructiveParody). We believe that this may be due to the large number of remakes and adaptations of other works that are currently being produced, so we speculate that many of new works will also be adapted to the new times in one way or another.

However, when applying the Leiden algorithm to the co-film network, fewer communities have been obtained, but they clearly differentiated with respect to genres. There is also a

significant difference between the rating and popularity of these communities. The outcomes of this research regarding the communities of tropes and films are useful for researchers and authors, as they can easily serve to categorize works in different levels of development, help on their creation and set expectations by simple comparisons. "Locking" on a genre or combination of genres, again, can be interpreted in several possible ways from the point of view of plot generation. Either you directly choose from the set of tropes that are essential to the genre, or you use genre-defying tropes drawn from the under-developed area of the troposphere, to create something with the potential to be popular. After all, popularity breeds itself, but it is also true that originality and novelty breeds popularity. At any rate, different environments might warrant different uses of the troposphere analysis: you might want to stick to the tried and true in the case of creating a backstory for a non-playing character in the videogame, you might want to mix that with using under-developed areas of the troposphere by putting together so-far untested combination of tropes to create things like a teen zombie musical movie (Anna and the Apocalypse (2018)).

It is important to note that although we use scientiometric-based techniques there are some differences from the analysis of scientific papers: the films analyzed have a very variable number of tropes, the tropes are more abstract in nature, the impact (popularity and rating) is subjective, and the source of the data (TVTropes) also receives varying degrees of attention as it is developed. This issue can be addressed by adding other sources of data such as RottenTomatoes (https://www.rottentomatoes.com/) or MetaCritic (http://www.metacritic.com). Furthermore, as in bibliometric analysis, the merely use a specific keyword (or trope) does not guarantee an increase in the impact of the work, but is indicative of the interest of the community to which they belong. In the case of films, it will also be necessary to take into consideration other elements, such as the actors, directors or funding, among others.

This same methodology can be applied to the same dataset, but at different levels, for example, to study the evolution of the tropes in a genre, country, or in a specific decade, as the study of tropes have importance to those scholars interested in film story [17]. Furthermore, TV Tropes also includes tropes in other media, such as videogames or comics, so a similar analysis could be also performed in those different fields. Moreover, researchers working on automatic story generation, and specifically those based on tropes, such as [5], can use this methodology to detect which tropes may be most suitable for their research.

## Author Contributions

**Conceptualization:** Pablo García-Sánchez, Juan Julián Merelo, Manuel Jesús Cobo.

**Formal analysis:** Pablo García-Sánchez, Antonio Velez-Estevez.

**Funding acquisition:** Juan Julián Merelo.

**Investigation:** Pablo García-Sánchez.

**Methodology:** Pablo García-Sánchez, Antonio Velez-Estevez.

**Supervision:** Manuel Jesús Cobo.

**Visualization:** Pablo García-Sánchez, Antonio Velez-Estevez.

**Writing – original draft:** Pablo García-Sánchez, Antonio Velez-Estevez.

**Writing – review & editing:** Juan Julián Merelo, Manuel Jesús Cobo.

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
