## [Decision Letter · Decision Letter 0]

12 Jan 2021

PONE-D-20-25471

The Simpsons did it: exploring the film trope space and its large scale structure

PLOS ONE

Dear Dr. García Sánchez,

Thank you for submitting your manuscript to PLOS ONE. After careful consideration, we feel that it has merit but does not fully meet PLOS ONE’s publication criteria as it currently stands. Therefore, we invite you to submit a revised version of the manuscript that addresses the points raised during the review process.

We look forward to receiving your revised manuscript.

Kind regards,

Chi-Hua Chen, Ph.D.

Academic Editor

PLOS ONE

Journal Requirements:

Reviewers' comments:

Reviewer's Responses to Questions

**Comments to the Author**

1. Is the manuscript technically sound, and do the data support the conclusions?

Reviewer #1: Yes

Reviewer #2: Yes

Reviewer #3: Partly

Reviewer #4: Yes

2. Has the statistical analysis been performed appropriately and rigorously? 

Reviewer #1: Yes

Reviewer #2: Yes

Reviewer #3: N/A

Reviewer #4: Yes

3. Have the authors made all data underlying the findings in their manuscript fully available?

Reviewer #1: Yes

Reviewer #2: Yes

Reviewer #3: Yes

Reviewer #4: Yes

4. Is the manuscript presented in an intelligible fashion and written in standard English?

Reviewer #1: No

Reviewer #2: Yes

Reviewer #3: Yes

Reviewer #4: Yes

5. Review Comments to the Author

Reviewer #1: This paper provides insights in understanding the tropes that compose the films. The authors study data from TV Tropes and IMDb websites using scientometric and complex networks methods.

The dataset is interesting and well described. The paper has a proper introduction into the research questions. However, there are a number of things regarding the description of methodology that could be improved. Below are comments that could help in making some sections more clear to a reader:

1. The authors often use "co-words network" and "co-occurence network" as metaphors to "co-tropes" or "co-films" network. This could be misleading and confusing, e.g. Algorithm 1, "Fig. 8 shows the spanning tree of the co-word network", ...

2. The authors use Jaccard index to compare tropes between two genres. The explanation "the Jaccard index is computed for each set of tropes within the genres in respect to the rest of genres tropes." is not precise.

3. The authors should define difference between community and cluster. Sometimes, it seems that authors use these two terms as synonyms. However, in l288, p8 they write: "This means that a film can belong to more than one cluster for the same community."

4. The authors should provide exact definition of density, centrality and central trope.

5. In Table 2, the authors should state that data for the period (2000,2020] is not complete.

6. In table 4 and 5, the authors should increase font size. Table 4 is not readable. Also, quality of the figures 5,6,8 and 9 is very poor and it is difficult to interpret the results.

7. The language needs to be improved since the manuscript contains a number of grammatical errors throughout it (the caption of Fig.1, etc).

Reviewer #2: In this interesting manuscript, authors analyse the distribution and association of tropes in movies, I.e. what they call troposphere. They do this by relying on network-based and scientometrics tools - which makes completely sense, as keywords in papers are like tropes in movies. The manuscript is really interesting and well written. To be honest, I have no major comments on it; for me, it could be published as it is. I’m just adding below a couple of “ideas” (please disregard them if not feasible), and some minor comments.

“Ideas”:

Evolution of trope reception. Authors have calculated the relationship between tropes and their reception, i.e. the associated films’ popularity. Yet, can it be that this association is temporal dependent? Imagine a film introducing a completely new trope, something never seen before. That movie could be appraised for being really original. Now, if hundreds of movies start using the same trope… that may get boring, and the associated rating may drop. Is there any way of measuring this?

(This last part reminds me of the YouTube channel CinemaSins… if you know it, you will surely understand why I’m suggesting this!)

Trope prediction. I would have enjoyed a final (speculative and personal, of course) prediction in the conclusions. For instance, having seen the data, which tropes are going to gain momentum in the future?

Minor comments:

- Line 138: “quality and popularity of the films”. Is quality a good word here? Because it suggests an objective measure, such as how good a film is from a technical perspective… but it’s not clear to me how this can be measured, and if this is actually taken into account in the manuscript.

- Line 257: “For the 100 most voted or rated tropes in genres.” Any rationale for choosing 100?

- Algorithm 1. Is this really needed? I mean, anyone working in networks will know how to create a co-ocurrence network… It seems to me that the textual explanation is more than enough.

- Line 280. “Clustering” may not be the best word here, as in complex network theory this reminds me of “assortativity”… why not just use modularity? Or “community detection”? At least, I would clarify that “clusters” and “communities” are used interchangeably in the paper, and that the former does not refer to transitivity.

- Line 517. “overview of which of these tropes may be interesting to develop in the future”. Not really, if I understood this correctly. That is, ascending and declining tropes are in the same quadrant… so, how can they be discriminated? With a time-dependent analysis?

- Some typos.

Line 219: “an specific methodology” -> “a”

Line 294: “To confirm that there is statistical significance between” I think “relationship” is missing… Same at line 427.

Line 550: “gender”? Or genre?

- On a side note, I find the abstract a little long. The third paragraph is perfect, but maybe the first two could be merged and eventually synthesised a little? In any case, this is more a question of style (and taste), not of scientific content.

Reviewer #3: This is an interesting paper studying the plot

of stories. While the results are relevant and worthy publishing,

some issues should be addrssed before this paper can be considered

adequate for publication:

- Section of related works could be improved by mentioning

similarities and differences with stories and texts modeled as

co-occurrence networks. See and mention e.g.

doi: 10.1209/0295-5075/114/58005 doi: 10.1016/J.INS.2018.02.047 .

- some steps in the methodology should be clarified.

For example, adjustments of the dataset is not informative.

This could mean anything.

- Methodological steps should be presented in a better way.

For example, performance analysis might include performance

measures and visualization. Alternatively, clustering analysis

could include items 6, 7 and 8.

- A descriptive analysis of the dataset could be provided

when presenting the dataset. This can not be considered a main result.

- It is not clear why Leiden algorithm have been used for this analysis.

I suggest using SBM to check the robustness of results.

- Could your methodology be a used in a practical contexts, e.g. to make any

type of prediction.

- It is not clear why word analysis mentioned here is related specifically

to bibliometrics analysis. co-occurrence analysis have been used in many contexts

other than scientometrics analysis, including e.g. in traditional text analysis:

doi: 10.1140/epjb/e2008-00206-x doi: 10.1016/j.physa.2018.03.013

- Figures are in low quality. I suggest using a vector format. I can not read most of labels in the larger network.

Reviewer #4: This paper investigates the nature of tropes in films using crowd-sourced data as its primary source material. The paper offers simple summary statistics of the data before applying statistical methods to develop network views in the spaces of tropes, genres and films.

First, a caveat. This reviewer is a physicist with a background in statistics of large-scale cosmic structure and no experience in reviewing Film, Television and Media (FTVM) literature. Like most adult humans, I do have some experience watching movies and the Simpsons.

The authors do not clear who is the primary audience for this work. Is this an academic work aimed at FTVM faculty? A practical work aimed at studio executives and/or independent filmmakers? An exploratory work aimed primarily at burnishing the reputations of the authors among a burgeoning group of trope analysts? The text is unclear.

In my opinion, the present draft has too many deficiencies to warrant publication as is. A revised version could be acceptable if it addresses these key, enumerated issues.

1) The tone of the work is too authoritative given its exploratory nature. For example, the abstract states that “Developing plots or narrations requires understanding the relationships between tropes in existing works,…” This is an overstatement; there is no such requirement. Line 11 of the main text claims we will “understand” human behaviors from crowdsourced data when it would be better to say that such sources offer a “window toward understanding”.

2) The figures and table are often difficult to read, particularly in printed form. Figures 6 and 8 are particular examples in which important labels are hard to make out. Tables 4 and 6 have very small font sizes.

3) The authors should do more to critically address the quality of their data. Should all of the nearly 26,000 tropes be considered as independent? Is popularity on IMdB different from that on other ratings platforms? The authors did not provide rationale for why the specific data sets (TV Tropes.org; IMDb) were used over, or in exclusion of, other databases such as allthetropes.com; Metacritic, Google User Reviews: Rotten Tomatoes. While the authors acknowledge bias in database used, cross referencing across multiple review databases would potentially produce different complex network results.

4) Language in the methodology section is awkward and contains grammatical errors that negatively affect readability. Would recommend editing for English clarity.

Overall I was hoping to learn something new from this paper but came away mostly confused. The authors would do well to embrace the “Less is more” philosophy here. A study confined to the top-100 most frequent tropes across the top 10 genres would be easier to digest compared to one that tackles everything all at once.

Other issues:

• Figures 1 and 3 would benefit from a log-log display (with special treatment for zeros), making the power-law nature of the frequency distributions more apparent.

• Lines 61,2: Sample biases must be understood to contextualize the understanding derived from network analysis.

• Line 80: Cite needed for Leiden algorithm.

• Line 106: “solve” -> “address”

• Line 114: “actual one” : how is the true trope nature of a film defined here?

• Line 122: who is the “us” being referred to here?

• Line 133: The community labels used here should be defined for the benefit of those unfamiliar with these terms. And/or a cite added.

• Lines 233,4,7: Examples of awkward English (Point 4 above).

• Line 281: A brief explanation of Jaccard index is warranted here. And/or move up explanatory equation.

• Lines 439-465: Table 4 is exasperating, nearly impossible to read. The description of the results here is helpful, but the fact that the Shout Out trope is nearly universal suggests that this trope should be considered separately from the others. Again, less could be more here.

• Line 513: grammar

• Line 550: gender -> genre

• Reference 1 is incomplete

6. PLOS authors have the option to publish the peer review history of their article (what does this mean?). If published, this will include your full peer review and any attached files.

Reviewer #1: No

Reviewer #2: **Yes: **Massimiliano Zanin

Reviewer #3: No

Reviewer #4: No

---

## [Author Response · Author response to Decision Letter 0]

22 Feb 2021

Dear Editor, 

We would like to express our gratitude to the editorial team and the reviewers, whose valuable comments and suggestions are highly appreciated and have been followed in this new version of the paper. The main differences with respect to the previous version are: 

- The abstract has been shortened.

- The motivation of the paper in the introduction has been clarified.

- New references have been added to improve the state of the art.

- Some concepts have been explained in more depth.

- Some decisions in the design of the methodology have been justified.

- Conclusions have been improved.

- The format of some figures has been changed and the font of the tables has been increased, as well as the correction of several grammatical errors.

Please find below a detailed description of the changes we have made in the manuscript to address the reviewers' comments. All modifications in the manuscript are highlighted in blue, including the changes after language correction.

Best regards.

The authors.

PS: This is the response to reviewers in text-only format as required by the system. The formated PDF with the sames responses in a nicer and more readable format is also attached.

Reviewer #1

REVIEWER INQUIRY

This paper provides insights in understanding the tropes that compose the films. The authors study data from TV Tropes and IMDb websites using scientometric and complex networks methods.

The dataset is interesting and well described. The paper has a proper introduction into the research questions. However, there are a number of things regarding the description of methodology that could be improved. Below are comments that could help in making some sections more clear to a reader

AUTHORS RESPONSE

We want to thank the reviewer for the kind comments.

REVIEWER INQUIRY

1. The authors often use "co-words network" and "co-occurence network" as metaphors to "co-tropes" or "co-films" network. This could be misleading and confusing, e.g. Algorithm 1, "Fig. 8 shows the spanning tree of the co-word network", ... 

AUTHORS RESPONSE

The reviewer is right. We have updated the references to ``co-word'' to ``co-trope'' or ``co-film'' instead, so that the document is easier to understand and better fits the network we are describing.

REVIEWER INQUIRY

2. The authors use Jaccard index to compare tropes between two genres. The explanation "the Jaccard index is computed for each set of tropes within the genres in respect to the rest of genres tropes." is not precise.

AUTHORS RESPONSE

We agree with the reviewer, the sentence is not precise and could lead to confusion. For each genre we gather the tropes that appear in the films of that genre. The Jaccard index is computed for each possible pair of two sets of tropes. Hence, to compare the genres, the Jaccard index is computed for each possible pair of set of tropes, each of them corresponds to a genre. To clarify this sentence, we have rewritten it as follows (line 295, page 7):

"""""

the Jaccard index is computed for each possible pair of set of tropes, each of them corresponding to a genre.

"""""

REVIEWER INQUIRY

3. The authors should define difference between community and cluster. Sometimes, it seems that authors use these two terms as synonyms. However, in l288, p8 they write: "This means that a film can belong to more than one cluster for the same community."

AUTHORS RESPONSE

We thank the reviewer for pointing this out. Indeed, community is a synonym for cluster in our manuscript. We should have written ``node'' instead of cluster. Hence, the sentence has been rewritten (line 333, page 9):

"""""

This means that a film can belong to more than one node for the same community.

"""""

Moreover, we have unified ``cluster" and ``community" in the rest of the text to avoid confusion.

REVIEWER INQUIRY

4. The authors should provide exact definition of density, centrality and central trope.

AUTHORS RESPONSE

We agree with the reviewer, therefore we have rewritten the visualization explanation including the definition of density and centrality. Moreover we have detailed the meaning of central trope and central film. The text is now written as follows (line 344, page 9):

"""""

 The strategic diagram is widely used in scientometrics, helping visualize the degree of development of a thematic community (cluster) by analyzing the networks of co-words using the keywords of the papers. It is based on two measures:

 - Callon's centrality: it indicates how well a theme is connected to other themes (external cohesion). It is defined as: $c=10\\times\\sum{e_{kh}}$ with $k$ word belonging to the theme and $h$ word belonging to another themes.

 - Callon's density: it measures the internal strength of a network and it can be defined as $d=100\\left(\\sum{e_{ij}/w}\\right)$, with $i$ and $j$ belonging to the theme and $w$ being the total number of words in the theme. 

 Moreover, the themes plotted in the strategic diagram are labelled by its central node (referred in the text as {central trope} or {central film}, depending on the network), which is the node with the most degree in the theme, formally $n=\\argmax_{v}\\left(\\deg(v)\\right)$. Using the callon's density and centrality it is possible to visualize the communities in four quadrants, whose center is in the position (0.5,0.5). These quadrants indicate if communities are motor (central and developed communities), transversal (basic and general), specialized/peripheral, or emerging/declining. This will help to answer RQ4.}

"""""

REVIEWER INQUIRY

5. In Table 2, the authors should state that data for the period (2000,2020] is not complete.

AUTHORS RESPONSE

Thanks to the reviewer's feedback we have included the statement in the caption of Table 2. Hence, the caption is written as follows:

"""""

Descriptive analysis of votes and ratings by periods for all films of the dataset. Notice that the data for the period (2000, 2020] is not complete.

"""""

REVIEWER INQUIRY

6. In table 4 and 5, the authors should increase font size. Table 4 is not readable. Also, quality of the figures 5,6,8 and 9 is very poor and it is difficult to interpret the results. 

AUTHORS RESPONSE

Font size in tables have been increased to facilitate the reading.

We are aware of the situation of the images, as other reviewers also pointed out the same issue. However, we have scrupulously followed the requirements of the article submission system: we uploaded our high-resolution TIFF images outside the text of the manuscript, using a web form in an additional step of the upload process. The images in the paper have been automatically generated at low resolution by the journal web system and embedded at the end of the PDF. We understand that in the final publication of the article these images will be displayed at higher resolution using our uploaded high-quality versions. We apologise for any inconvenience this may have caused.

REVIEWER INQUIRY

7. The language needs to be improved since the manuscript contains a number of grammatical errors throughout it (the caption of Fig.1, etc).

AUTHORS RESPONSE

We want to thank the reviewer suggestion. We have improved the language along the manuscript. Changes have been highlighted in blue.

Reviewer #2

REVIEWER INQUIRY

In this interesting manuscript, authors analyse the distribution and association of tropes in movies, I.e. what they call troposphere. They do this by relying on network-based and scientometrics tools - which makes completely sense, as keywords in papers are like tropes in movies. The manuscript is really interesting and well written. To be honest, I have no major comments on it; for me, it could be published as it is. I’m just adding below a couple of “ideas” (please disregard them if not feasible), and some minor comments. 

We thank the reviewer for the kind comments and are glad the reviewer found the paper interesting.

REVIEWER INQUIRY

“Ideas”:

Evolution of trope reception. Authors have calculated the relationship between tropes and their reception, i.e. the associated films’ popularity. Yet, can it be that this association is temporal dependent? Imagine a film introducing a completely new trope, something never seen before. That movie could be appraised for being really original. Now, if hundreds of movies start using the same trope… that may get boring, and the associated rating may drop. Is there any way of measuring this?

(This last part reminds me of the YouTube channel CinemaSins... if you know it, you will surely understand why I’m suggesting this!) 

AUTHORS RESPONSE

The reviewer is right. The slasher trope, and its first-person perspective, of the film Halloween had never been seen before and that is why this film is so good, even though such tropes are very common these days. We think it is hard to judge when a new trope will produce an improvement. For example, as a viewer's film culture increases, the appearance of repeated tropes can cause to lose interest in a movie they see later. Also, when a novel trope appears in place of a familiar one, it will not always be accepted by the audience. For example, in The Last Jedi, several tropes were subverted, which led to the rage of many fans. Therefore, it is difficult to gauge when the appearance (or subversion) of a new trope will be better than using the most common trope. Nevertheless, it is another very interesting topic to study in future work, and we thanks the reviewer for the idea.

By the way, if the reviewer is interested in the CinemaSins channel, we also recommend its not so well known nemesis: CinemaWins, which, on the contrary, shows the good parts of films, even from the movies that are reviled by critics.

REVIEWER INQUIRY

Trope prediction. I would have enjoyed a final (speculative and personal, of course) prediction in the conclusions. For instance, having seen the data, which tropes are going to gain momentum in the future? 

AUTHORS RESPONSE

This is a good idea. We have added the next paragraph in the conclusions (line 706, page 17).

"""""

As indicated in the results, there are communities in the emerging quadrant that are precisely related to previous adaptations or versions (ItWasHisSled, AdaptationalJerkass and DeconstructiveParody). We believe that this may be due to the large number of remakes and adaptations of other works that are currently being produced, so we speculate that many of new works will also be adapted to the new times in one way or another.

"""""

REVIEWER INQUIRY

Line 138: “quality and popularity of the films”. Is quality a good word here? Because it suggests an objective measure, such as how good a film is from a technical perspective... but it’s not clear to me how this can be measured, and if this is actually taken into account in the manuscript.

AUTHORS RESPONSE

We share the point of view of the reviewer. We have changed the word ``quality'' and used ``rating'' instead.

REVIEWER INQUIRY

Line 257: “For the 100 most voted or rated tropes in genres.” Any rationale for choosing 100? 

AUTHORS RESPONSE

We thank the reviewer question. Given the space of the paper, we have considered that 100 is an optimum number to show the most featured tropes. 

REVIEWER INQUIRY

Algorithm 1. Is this really needed? I mean, anyone working in networks will know how to create a co-ocurrence network… It seems to me that the textual explanation is more than enough.

AUTHORS RESPONSE

We appreciate the reviewer question. We took into account the potential public of this paper and we considered to include it for the sake of understandability.

REVIEWER INQUIRY

Line 280. “Clustering” may not be the best word here, as in complex network theory this reminds me of “assortativity”... why not just use modularity? Or “community detection”? At least, I would clarify that “clusters” and “communities” are used interchangeably in the paper, and that the former does not refer to transitivity.

AUTHORS RESPONSE

We agree with the reviewer. We have changed "Clustering" (line 280) by ``Community detection'' (now in line 324, page 8) which is more appropiate in the context. Moreover, we have unified the word ``cluster" into ``community" in the rest of the text to avoid potential confusions.

REVIEWER INQUIRY

Line 517. “overview of which of these tropes may be interesting to develop in the future”. Not really, if I understood this correctly. That is, ascending and declining tropes are in the same quadrant… so, how can they be discriminated? With a time-dependent analysis?

AUTHORS RESPONSE

The reviewer is right, the ascending and declining tropes are in the same quadrant, so they are discriminated via a time-dependent analysis as proposed in [3]. In this paper, we do it as shown in Figure 7 of the paper, where all the communities in the quadrant are ascending as the number of films increases by each decade. However, some communities are more noticeable than others such as ItWasHisSled or DeconstructiveParody.

REVIEWER INQUIRY

Line 219: “an specific methodology” \\verb|->| “a” 

Line 294: “To confirm that there is statistical significance between” I think “relationship” is missing… Same at line 427. 

Line 550: “gender”? Or genre?

AUTHORS RESPONSE

Thank the reviewer for the corrections, the proposed changes have been made.

REVIEWER INQUIRY

On a side note, I find the abstract a little long. The third paragraph is perfect, but maybe the first two could be merged and eventually synthesised a little? In any case, this is more a question of style (and taste), not of scientific content.

AUTHORS RESPONSE

Without losing information, these two paragraphs have been merged and now read:

""""

 Creating a story is a challenging task due to the the complex relations between the parts that make it up, which is why many new stories are built on those cohesive elements or patterns, called {tropes} that have been shown to work in the past. A tropes is a recurring storytelling device or pattern, or sometimes meta-elements, used by the authors to express ideas that the audience can recognize or relate to, such as the {Hero's Journey}. Discovering tropes and how they cluster in popular works and doing it at scale to generate new plots may benefit writers. In this paper, we analyze them and use a principled procedure to identify trope combinations, or communities, that could possible be successful. The degree of development of these different communities can help us identify areas that are under-developed and, thus, susceptible to such a type of development.

""""

We thank the reviewer for this suggestion.

Reviewer #3

REVIEWER INQUIRY

This is an interesting paper studying the plot

of stories. While the results are relevant and worthy publishing,

some issues should be addrssed before this paper can be considered

adequate for publication:

AUTHORS RESPONSE

We want to thank the reviewer for the and we are very glad that you found the paper interesting.

REVIEWER INQUIRY

Section of related works could be improved by mentioning

similarities and differences with stories and texts modeled as

co-occurrence networks. See and mention e.g.

doi: 10.1209/0295-5075/114/58005 doi: 10.1016/J.INS.2018.02.047 .

AUTHORS RESPONSE

We thank the reviewer for these new references. Both of them have been added and commented in the state of the art (line 222, page 6):

"""""

Amancio [2] proposed the use of co-occurrence networks to study and detect the different entities (characters or locations) and their semantic relationships, which appear in different novels. A topological analysis was applied to obtain patterns that were not possible to detect using classical methods. Another application of the use of complex networks is the identification of the meaning of words with multiple meanings. The work [10] describes a process based on a bi-partite network model that outperforms widely used machine learning methods to deal with this issue. However, as stated before, these works rely on the analysis of whole texts to obtain the relationships between entities, whereas our work is based on the mere occurrence of tropes in the different films.

"""""

REVIEWER INQUIRY

some steps in the methodology should be clarified.

For example, adjustments of the dataset is not informative.

This could mean anything.

AUTHORS RESPONSE

We agree with the reviewer, we have changed the name to ``Dataset preprocessing''.

REVIEWER INQUIRY

Methodological steps should be presented in a better way.

For example, performance analysis might include performance

measures and visualization. Alternatively, clustering analysis

could include items 6, 7 and 8.

AUTHORS RESPONSE

We want to thank the reviewer for the comment. The steps explained in this manuscript, are described in a similar way in other published works [1]. Nonetheless, some names may lead to confusion so we have changed them and modified the ``Adjustments of the dataset" step as follows (line 277, page 7):

"""Dataset pre-processing: in this study, two datasets are considered:"""

"""The dataset obtained from the data acquisition phase. In the rest of the text it will be referred as $D_f$."""

REVIEWER INQUIRY

A descriptive analysis of the dataset could be provided

when presenting the dataset. This can not be considered a main result.

AUTHORS RESPONSE

Thank you for your comment. We present the descriptive analysis in a different section, Experiments and results, and it is used to solve the first research question (RQ1).

REVIEWER INQUIRY

It is not clear why Leiden algorithm have been used for this analysis.

I suggest using SBM to check the robustness of results.

AUTHORS RESPONSE

Thank you for your comment. In the literature [5], various algorithms have been proposed to deal with community detection in networks, in fact, methods such as blockmodeling, spectral clustering, or modularity-based clustering among others are very adequate to address the problem. It is also known that there is no algorithm better than other but it depends on the needs of the study. In our case we have used the Leiden algorithm because its features suit perfectly with the problem we are addressing. The features that lead us to choose the Leiden algorithm are:

 - It ensures well connected communities.

 - All the communities are subset optimal, which ensures that all subsets of the community are locally optimally assigned and cannot be moved to a different community.

According to the above arguments, we think that Leiden algorithm, which is modularity-based, is a good option but other algorithms would be valid as well. 

Concerning the robustness, we know that algorithms based on modularity could give different results. In our case, we have executed of the Leiden algorithm with the \\verb|n_iterations| input parameter equals \\verb|-1|, so it is run until an iteration in which there is no improvement. Therefore, we have checked the robustness of our results in two different ways:

-We ran the algorithm multiple times and no substantial differences in the communities were found.

- The detection of communities have been examined by domain experts, who agree with the results.

In addition, we want to point out that the Leiden algorithm [14], published on the journal Scientific Reports (Nature Publishing Group) in 2019, has 94 citations in the Web of Science database, and among them, there are important applications to other fields such as:

 - ``Antibiotics-Driven Gut Microbiome Perturbation Alters Immunity to

 Vaccines in Humans" [8], published on Cell journal, which is the first journal in terms of impact factor in the area of Biochemistry \\& Molecular Biology, as well as the second one in the area of Cell Biology.

 - ``Identification of a T follicular helper cell subset that drives anaphylactic IgE" [7], published on Science journal, which is the second journal ranked in Multidisciplinary Sciences.

 - ``Assessing police topological eficiency in a major sting operation on the dark web" [4], published on Scientific Reports, which has a great impact factor in the area of Multidisciplinary Sciences.

Furthermore, the Centre for Science and Technology Studies (CWTS) based in Leiden, has collaborated in the development of Citation Topics in InCites (Clarivate) [11]. Citation Topics is a new document-level classification scheme for InCites Benchmarking \\& Analytics™, and it has been developed with the power of the Leiden community detection algorithm as commented in the Global ISI Reports 2021 [12]. 

REVIEWER INQUIRY

Could your methodology be a used in a practical contexts, e.g. to make any type of prediction.

AUTHORS RESPONSE

The practical applications of this paper may be of interest to a wide range of professionals. For example, film historians can use our methodology to study how tropes have evolved over time, or the cultural differences from films of different countries. Moreover, understanding how tropes relate to each other in a quantitative way can also benefit researchers in other fields, such as automatic narrative generation. We have therefore added the following paragraph to the paper (line 167, page 5):

"""

In general, we think that this methodology can be useful since it gives us a quantitative approach to the analysis of narrative devices, which can be easily turned into a methodology for generation of narratives, as a well as a for critical analysis of these systematized narratives in the past. In this sense, we think it can benefit videogame creators, as well as media history scholars.}

"""

REVIEWER INQUIRY

It is not clear why word analysis mentioned here is related specifically

to bibliometrics analysis. co-occurrence analysis have been used in many contexts

other than scientometrics analysis, including e.g. in traditional text analysis:

doi: 10.1140/epjb/e2008-00206-x doi: 10.1016/j.physa.2018.03.013 \\\\

AUTHORS RESPONSE

The reviewer is right. We related word analysis specifically to bibliometrics analysis because we have a bibliometrics background. Nonetheless, we agree with the comment, and included the references given by the reviewer in the related work as follows (line 209, page 5):

"""""

Moreover, word analysis has been applied in other contexts such as traditional text analysis. For instance, Herrera et al. [9] proposed a detector, which is based on unsupervised statistical methods, for detecting keywords in texts. Other application to traditional text analysis is presented by Tohalino et al. [13], in which a multilayer network is used to address the extractive multidocument summarization task. However, in our study we do not analyze corpus of text, but the relationship between movies and the tropes they appear in them, and vice versa.

}"""""

REVIEWER INQUIRY

Figures are in low quality. I suggest using a vector format. I can not read most of labels in the larger network. 

AUTHORS RESPONSE

We are aware of the situation, as other reviewers also pointed out the same issue. However, we have scrupulously followed the requirements of the article submission system: we uploaded our high-resolution TIFF images outside the text of the manuscript, using a web form in an additional step of the upload process. The images in the paper have been automatically generated at low resolution by the journal web system and embedded at the end of the PDF. We understand that in the final publication of the article these images will be displayed at higher resolution using our uploaded high-quality versions. We apologise for any inconvenience this may have caused.

Reviewer #4} 

REVIEWER INQUIRY

This paper investigates the nature of tropes in films using crowd-sourced data as its primary source material. The paper offers simple summary statistics of the data before applying statistical methods to develop network views in the spaces of tropes, genres and films.

First, a caveat. This reviewer is a physicist with a background in statistics of large-scale cosmic structure and no experience in reviewing Film, Television and Media (FTVM) literature. Like most adult humans, I do have some experience watching movies and the Simpsons.

AUTHORS RESPONSE

We want to thank the reviewer for the valuable comments.

REVIEWER INQUIRY

The authors do not clear who is the primary audience for this work. Is this an academic work aimed at FTVM faculty? A practical work aimed at studio executives and/or independent filmmakers? An exploratory work aimed primarily at burnishing the reputations of the authors among a burgeoning group of trope analysts? The text is unclear.

In my opinion, the present draft has too many deficiencies to warrant publication as is. A revised version could be acceptable if it addresses these key, enumerated issues.

AUTHORS RESPONSE

Studio executives and/or independent filmmakers are not known to read scholarly papers. We'd be much honored if they did, but they will probably not. This is, quite obviously, an academic work, aimed at the readers of this journal; it's going to be openly published, so its audience might potentially quite wide. We really don't know there's such thing as a group of trope analysts, but we certainly hope that if the paper is published (with your approval), our reputation is not tarnished.

However, I think the specific audience is beside the point of the paper (and it should certainly be for most papers we have written in the past and will write in the future). The objective of the paper is to provide a way to analyze story patterns at scale, using as source a crowdsourced platform. As such, it can benefit historians, the FTVM faculty you mention, as well as anyone wanting to generate stories at scale, such as video game producers. You certainly have a valid point indicating that this has not been sufficiently clarified in the paper, which is why we have added this (line 167, page 5):

"""

 In general, we think that this methodology can be useful since it gives us a quantitative approach to the analysis of narrative devices, which can be easily turned into a methodology for generation of narratives, as a well as a for critical analysis of these systematized narratives in the past. In this sense, we think it can benefit video game creators, as well as media history scholars.}

"""

We will also do our best to address your other points below.

REVIEWER INQUIRY

1) The tone of the work is too authoritative given its exploratory nature. For example, the abstract states that “Developing plots or narrations requires understanding the relationships between tropes in existing works,…” This is an overstatement; there is no such requirement. Line 11 of the main text claims we will “understand” human behaviors from crowdsourced data when it would be better to say that such sources offer a “window toward understanding”.

AUTHORS RESPONSE

The first two paragraphs of the abstract have been rewritten, they now read

"""

 Creating a story is a challenging task due to the the complex relations between the parts that make it up, which is why many new stories are built on those cohesive elements or patterns, called {tropes} that have been shown to work in the past. A trope is a recurring storytelling device or pattern, or sometimes a meta-element, used by the authors to express ideas that the audience can recognize or relate to, such as the {Hero's Journey}. Discovering tropes and how they cluster in popular works and doing it at scale to generate new plots may benefit writers; in this paper, we analyze them and use a principled procedure to identify trope combinations, or communities, that could possible be successful. The degree of development of these different communities can help us identify areas that are under-developed and, thus, susceptible to such a type of development.}

"""

The second request has been changed in the way requested. Thanks for the suggestion.

REVIEWER INQUIRY

2) The figures and table are often difficult to read, particularly in printed form. Figures 6 and 8 are particular examples in which important labels are hard to make out. Tables 4 and 6 have very small font sizes. 

AUTHORS RESPONSE

Font size in tables have been increased.

We are aware of the situation of the Figures, as other reviewers also pointed out the same issue. However, we have scrupulously followed the requirements of the article submission system: we uploaded our high-resolution TIFF images outside the text of the manuscript, using a web form in an additional step of the upload process. The images in the paper have been automatically generated at low resolution by the journal web system and embedded at the end of the PDF. We understand that in the final publication of the article these images will be displayed at higher resolution using our uploaded high-quality versions. We apologise for any inconvenience this may have caused.

REVIEWER INQUIRY

3) The authors should do more to critically address the quality of their data. Should all of the nearly 26,000 tropes be considered as independent? Is popularity on IMdB different from that on other ratings platforms? The authors did not provide rationale for why the specific data sets (TV Tropes.org; IMDb) were used over, or in exclusion of, other databases such as allthetropes.com; Metacritic, Google User Reviews: Rotten Tomatoes. While the authors acknowledge bias in database used, cross referencing across multiple review databases would potentially produce different complex network results. 

AUTHORS RESPONSE

We have used IMDB because it allows you to use an API in a straightforward way and get batch information for all their films at once. In the case of RottenTomatoes and MetaCritic, requests have to be made individually per film, and there is also a limit per user, making the download of all the data unfeasible in a reasonable time. In addition, the metadata of the retrieved films is more limited. 

Regarding the downloading of tropes, we have used TVTropes instead of AllTheTropes, because the latter is a fork of the former, containing practically the same information, but TVTropes being more widely used and known.

Nevertheless, the idea proposed by the reviewer is interesting for future studies, so it has been mentioned in future work at the end of the paper (line 733, page 18).

"""""

\\{This issue can be addressed by adding other sources of data such as RottenTomatoes\\footnote{\\url{https://www.rottentomatoes.com/}} or MetaCritic\\footnote{\\url{http://www.metacritic.com}}.}

"""""

REVIEWER INQUIRY

4) Language in the methodology section is awkward and contains grammatical errors that negatively affect readability. Would recommend editing for English clarity.

AUTHORS RESPONSE

Thank to the reviewer comment. We have improved the language of the paper.

REVIEWER INQUIRY

Overall I was hoping to learn something new from this paper but came away mostly confused. The authors would do well to embrace the “Less is more” philosophy here. A study confined to the top-100 most frequent tropes across the top 10 genres would be easier to digest compared to one that tackles everything all at once.

AUTHORS RESPONSE

We understand what the reviewer is suggesting, and hope this new version is less confusing than the initial one, after addressing all concerns. However, we would like to point out that the paper is mainly methodological and is not so much concerned with creating (forcibly ephemeral) rankings, than with proposing a methodology to analyze how tropes cluster together in movies, and how this can be used as an inspiration for future naration topics.

We still think that the paper works as a whole, since we are trying to address different research questions and all parts work together towards the answer. We will try, however, to be if not briefer, at least more direct in getting through to the prospective reader. We will try to address the rest of your concerns below.

REVIEWER INQUIRY

Figures 1 and 3 would benefit from a log-log display (with special treatment for zeros), making the power-law nature of the frequency distributions more apparent.

AUTHORS RESPONSE

We appreciate the reviewer suggestion. We have updated the figures to the requested log-log display, they are shown in Figures 1 and 3. Moreover, we have rewritten the captions to improve the awkward English as other reviewers have pointed out. Additionally, we have referred to the Figures 1 and 3 as log-normal distributions.

REVIEWER INQUIRY

Lines 61,2: Sample biases must be understood to contextualize the understanding derived from network analysis.

AUTHORS RESPONSE

As indicated in [6], there is clearly a bias in the sample that we are using: more popular movies will attract more attention from the public, and more tropes will be reported, and and the time bias is also clear, with current movies getting more tropes than old ones. This is inherent in the source we are using, and there's relatively little we can do to overcome this. We have added, however, a clarification about this bias, clarifying that, while it would obviously be a problem in a rigorous and precise historic and quantitative analysis, this is not really our objective. This text has been added to the paper to clarify this point (line 72, page 3).

"""

In this endeavor we should not lose sight of the fact that the source that we are using is biased towards popular and recent films, so some stories that have not reached popularity for some reason (minority language, for instance) or that were released a long time ago might have a winning trope combination that is not reflected in our study. Besides, this bias is not constant, so we can't really affirm that some specific genre or kind of movie has been left out (or included) uniformly across the data we have analyzed.

Fortunately, this bias aligns with our objective, which is narrative generation in the present sense, as well as the creation of narratives that might be cohesive. We will assume that a part of what makes a movie popular is the cohesiveness of the tropes they employ, or how they work together with each other. This means that our results will probably not be affected by this bias, that will nonetheless have to be taken into account when working towards a different objective.

"""

REVIEWER INQUIRY

Line 80: Cite needed for Leiden algorithm.

AUTHORS RESPONSE

As the reviewer pointed out, we included the cite.

REVIEWER INQUIRY

Line 114: “actual one” : how is the true trope nature of a film defined here?

AUTHORS RESPONSE

We have rewritten the phrase, as we were referring to the actual set of tropes in a film, with respect to the set detected by users on the website.

REVIEWER INQUIRY

Line 133: The community labels used here should be defined for the benefit of those unfamiliar with these terms. And/or a cite added.

AUTHORS RESPONSE

Thank you for your comment. We have cited the paper where the community labels are explained.

REVIEWER INQUIRY

Line 281: A brief explanation of Jaccard index is warranted here. And/or move up explanatory equation.

AUTHORS RESPONSE

Thank you for your comment. We have included a brief explanation of Jaccard index in the overlapping analysis step of the methodology. The included text is the following (line 292, page 7):

"""""

A well-known measure of overlapping is the Jaccard index, which measures the similarity between two sets, and it is defined as the size of the intersection divided by the size of the union of the sets given.

"""""

REVIEWER INQUIRY

 Lines 439-465: Table 4 is exasperating, nearly impossible to read. The description of the results here is helpful, but the fact that the Shout Out trope is nearly universal suggests that this trope should be considered separately from the others. Again, less could be more here.

AUTHORS RESPONSE

Reviewer is right. We have increased the font size for these tables. Quite clearly, ShoutOut, which is essentially a reference to other films, characters or popular culture, is a staple of modern films, and it really stands out among the rest. However, our methodology does not allow to consider it separately. It would be interesting, however, to study how the inherently networked nature of this trope makes it so popular, and literally so central. That is, however, beyond the focus of this paper. 

REVIEWER INQUIRY

Line 513: grammar 

Lines 233,4,7: Examples of awkward English (Point 4 above). 

Line 122: who is the “us” being referred to here? 

Line 106: “solve” \\verb|->| “address” 

Line 550: gender \\verb|->| genre 

AUTHORS RESPONSE

We changed all the minor corrections proposed by the reviewer. Regarding the awkward English, we have done the following:

 Phrase in Line 513 has been rewritten (now in line 565, page 14):

 """""

 In addition, there has been an apparent increase in the number of films from some communities in recent years.}

 """""

 Lines 233-234. The English has been reviewed and we have extended the explanation for the sake of clarity. Therefore the paragraph has been changed to (now in line 267, page 7):

 """""

 Preliminary analysis of the dataset: To get an overview of the dataset, a descriptive analysis is proposed. The analysis includes the standard deviation, mean, minimum, maximum and the first, second and third quartiles of votes, ratings and tropes. Moreover, a visualization of the distributions of the number of tropes, ratings and votes is shown. Furthermore, a line chart with the films per year is presented. Finally, the votes and ratings in periods of two decades from 1880 to 2020 are analysed as well as the distribution of genres in the dataset.

All the previous analysis will help us to detect whether some biases exist or not. Additionally, the analysis will give information to answer the first research question (RQ1).

 """""

 Line 237. We have changed the sentence to (now in line 278, page 7): 

 """""

 The dataset obtained from the data acquisition phase. In the rest of the text it will be referred as $D_f$.

 """""

Moreover, concerning the line 122, we removed the ``us'' so the sentence now is written as follows (line 134, page 4):

"""""

generally genre (or combination thereof) comes before the plot itself, and understanding how tropes shape genres or the other way out will help to generate better stories.

"""""

REVIEWER INQUIRY

Reference 1 is incomplete 

AUTHORS RESPONSE

Thanks to the reviewer's feedback, we updated the reference 1. Now, it is written as follows:

"""""

Bellanova R, González Fuster G. No (Big) Data, no fiction? Thinking surveillance with/against Netflix. In: Rudinow A, Schneider I, Green N, ors. The Politics and Policies of Big Data: Big Data Big Brother? Routledge; 2018. p. 227–246.

"""""

REFERENCES

[1] Laura Alcaide–Muñoz, Manuel Pedro Rodriguez–Bolivar, Manuel Jesús

Cobo, and Enrique Herrera–Viedma. Analysing the scientific evolution of

e-government using a science mapping approach. Government Information

Quarterly, 34(3):545 – 555, 2017.

[2] Diego Raphael Amancio. Network analysis of named entity co-occurrences in written texts. EPL (Europhysics Letters), 114(5):58005, jun 2016.

[3] M J Cobo, A G López-Herrera, E Herrera-Viedma, and F Herrera. An

approach for detecting, quantifying, and visualizing the evolution of a research field: A practical application to the Fuzzy Sets Theory field. Journal of Informetrics, 5(1):146–166, 2011.

[4] Bruno Requiao da Cunha, Padraig MacCarron, Jean Fernando Passold,

Luiz Walmocyr dos Santos, Jr., Kleber A. Oliveira, and James P. Gleeson.

Assessing police topological efficiency in a major sting operation on the dark web. SCIENTIFIC REPORTS, 10(1), JAN 9 2020.

[5] Santo Fortunato. Community detection in graphs.

486(3):75 – 174, 2010. Physics Reports,

[6] Rubén Héctor Garcia-Ortega, Pablo Garcia-Sánchez, and Juan

Julián Merelo Guervós. Tropes in films: an initial analysis. CoRR,

abs/2006.05380, 2020.

[7] Uthaman Gowthaman, Jennifer S. Chen, Biyan Zhang, William F. Flynn,

Yisi Lu, Wenzhi Song, Julie Joseph, Jake A. Gertie, Lan Xu, Magalie A.

Collet, Jessica D. S. Grassmann, Tregony Simoneau, David Chiang, M. Ce-

cilia Berin, Joseph E. Craft, Jason S. Weinstein, Adam Williams, and

Stephanie C. Eisenbarth. Identification of a T follicular helper cell subset that drives anaphylactic IgE. SCIENCE, 365(6456):883+, AUG 30 2019.

[8] Thomas Hagan, Mario Cortese, Nadine Rouphael, Carolyn Boudreau,

Caitlin Linde, Mohan S. Maddur, Jishnu Das, Hong Wang, Jenna Guth-

miller, Nai-Ying Zheng, Min Huang, Amit A. Uphadhyay, Luiz Gardinassi,

Caroline Petitdemange, Michele Paine McCullough, Sara Jo Johnson, Kiran

Gill, Barbara Cervasi, Jun Zou, Alexis Bretin, Megan Hahn, Andrew T.

Gewirtz, Steve E. Bosinger, Patrick C. Wilson, Shuzhao Li, Galit Alter,

Surender Khurana, Hana Golding, and Bali Pulendran. Antibiotics-Driven

Gut Microbiome Perturbation Alters Immunity to Vaccines in Humans.

CELL, 178(6):1313+, SEP 5 2019.

[9] J. P. Herrera and P. A. Pury. Statistical keyword detection in literary corpora. European Physical Journal B, 2008.

[10] Edilson Anselmo Corrêa Júnior, Alneu de Andrade Lopes, and Diego R.

Amancio. Word sense disambiguation: A complex network approach. Inf.

Sci., 442-443:103–113, 2018.

[11] Ingo Lütkebohle. Introducing Citation Topics in In-

Cites. https://clarivate.com/webofsciencegroup/article/

introducing-citation-topics/, 2020. [Online; accessed 03-02-2021].

[12] Martin Szomszor, Jonathan Adams, David A. Pendlebury, and

Gordon Rogers.

Data categorization: understanding choices and

outcomes.

https://clarivate.com/webofsciencegroup/campaigns/

data-categorization-understanding-choices-and-outcomes/, 2021.

[Online; accessed 17-02-2021].

[13] Jorge V. Tohalino and Diego R. Amancio. Extractive multi-document summarization using multilayer networks. Physica A: Statistical Mechanics and its Applications, 2018.

[14] V. A. Traag, L. Waltman, and N. J. van Eck. From Louvain to Leiden:

guaranteeing well-connected communities. Scientific Reports, 9(1):5233,

2019.

---

## [Decision Letter · Decision Letter 1]

8 Mar 2021

The Simpsons did it: exploring the film trope space and its large scale structure

PONE-D-20-25471R1

Dear Dr. García Sánchez,

We’re pleased to inform you that your manuscript has been judged scientifically suitable for publication and will be formally accepted for publication once it meets all outstanding technical requirements.

Kind regards,

Chi-Hua Chen, Ph.D.

Academic Editor

PLOS ONE

Additional Editor Comments (optional):

Reviewers' comments:

Reviewer's Responses to Questions

**Comments to the Author**

1. If the authors have adequately addressed your comments raised in a previous round of review and you feel that this manuscript is now acceptable for publication, you may indicate that here to bypass the “Comments to the Author” section, enter your conflict of interest statement in the “Confidential to Editor” section, and submit your "Accept" recommendation.

Reviewer #1: All comments have been addressed

Reviewer #2: All comments have been addressed

Reviewer #3: All comments have been addressed

2. Is the manuscript technically sound, and do the data support the conclusions?

Reviewer #1: Yes

Reviewer #2: Yes

Reviewer #3: Yes

3. Has the statistical analysis been performed appropriately and rigorously? 

Reviewer #1: Yes

Reviewer #2: Yes

Reviewer #3: Yes

4. Have the authors made all data underlying the findings in their manuscript fully available?

Reviewer #1: Yes

Reviewer #2: Yes

Reviewer #3: Yes

5. Is the manuscript presented in an intelligible fashion and written in standard English?

Reviewer #1: Yes

Reviewer #2: Yes

Reviewer #3: Yes

6. Review Comments to the Author

Reviewer #1: The authors have adequately addressed the comments and I can recommend the manuscript for publication. However, I suggest the authors to carefully read the manuscript and correct grammatical errors.

Reviewer #2: Authors have addressed all my comments, and the initial version of the manuscript was already good, so I'm OK with this version.

Reviewer #3: All issues have been addressed by the author.

The authors have clarified my previous concerns.

I recommend this paper to be accepted.

7. PLOS authors have the option to publish the peer review history of their article (what does this mean?). If published, this will include your full peer review and any attached files.

Reviewer #1: No

Reviewer #2: **Yes: **Massimiliano Zanin

Reviewer #3: No

---

## [Editor Report · Acceptance letter]

10 Mar 2021

PONE-D-20-25471R1 

The Simpsons did it: exploring the film trope space and its large scale structure  

Dear Dr. García Sánchez:

I'm pleased to inform you that your manuscript has been deemed suitable for publication in PLOS ONE. Congratulations! Your manuscript is now with our production department. 

Kind regards, 

on behalf of

Professor Chi-Hua Chen 

Academic Editor

PLOS ONE